# Specific structural elements of the T-box riboswitch drive the two-step binding of the tRNA ligand

Jiacheng Zhang[1†], Bhaskar Chetnani[2†‡], Eric D Cormack[3], Dulce Alonso[2], Wei Liu[4], Alfonso Mondragón[2], Jingyi Fei[1,4*]

[1]Institute for Biophysical Dynamics, University of Chicago, Chicago, United States; [2]Department of Molecular Biosciences, Northwestern University, Evanston, United States; [3]University of Chicago College, Chicago, United States; [4]Department of Biochemistry and Molecular Biology, University of Chicago, Chicago, United States

**Abstract** T-box riboswitches are *cis*-regulatory RNA elements that regulate the expression of proteins involved in amino acid biosynthesis and transport by binding to specific tRNAs and sensing their aminoacylation state. While the T-box modular structural elements that recognize different parts of a tRNA have been identified, the kinetic trajectory describing how these interactions are established temporally remains unclear. Using smFRET, we demonstrate that tRNA binds to the riboswitch in two steps, first anticodon recognition followed by the sensing of the 3' NCCA end, with the second step accompanied by a T-box riboswitch conformational change. Studies on site-specific mutants highlight that specific T-box structural elements drive the two-step binding process in a modular fashion. Our results set up a kinetic framework describing tRNA binding by T-box riboswitches, and suggest such binding mechanism is kinetically beneficial for efficient, co-transcriptional recognition of the cognate tRNA ligand.
DOI: https://doi.org/10.7554/eLife.39518.001

*For correspondence:
jingyifei@uchicago.edu

†These authors contributed equally to this work

Present address: ‡Innovation and Technology Transfer Division, Rush University Medical Center, Chicago, United States

Competing interests: The authors declare that no competing interests exist.

## Introduction

Riboswitches are *cis*-regulatory RNA elements that recognize and respond to defined external signals to affect transcription or translation of downstream messenger RNAs (mRNAs) (*Breaker, 2012*; *Serganov and Nudler, 2013*; *Sherwood and Henkin, 2016*). Riboswitches generally consist of two domains: a sensory or aptamer domain and a regulatory domain or expression platform. The expression platform can adopt different conformations in response to ligand binding to the aptamer, and in this way control gene expression outcome (*Breaker, 2012*). The aptamer of each riboswitch class contains conserved sequence motifs and unique secondary or tertiary structural elements that help distinguish and bind specific ligands (*McCown et al., 2017*). Bacterial T-box riboswitches represent a unique class of riboswitches that do not bind small molecule ligands, instead they recognize and bind tRNA molecules and sense directly their aminoacylation state (*Zhang and Ferré-D'Amaré, 2015*). T-box riboswitches serve as excellent paradigms to understand RNA-RNA interactions and RNA-based regulation.

T-box riboswitches are found in Gram-positive bacteria and are usually located in the region upstream of mRNA sequences encoding aminoacyl tRNA synthetases and proteins involved in amino acid biosynthesis and transport and hence participate directly in amino acid homeostasis (*Zhang and Ferré-D'Amaré, 2015*). In general, the aptamer domain of all T-box riboswitches contains a long stem, Stem I, responsible for specific tRNA binding (*Rollins et al., 1997*). The expression platform can adopt either a terminator or anti-terminator conformation, depending on whether the bound tRNA is charged or uncharged (*Henkin, 2014*; *Zhang and Ferré-D'Amaré, 2015*). In most T-box

**eLife digest** Living organisms depend upon a group of chemicals called amino acids to survive. Amino acids are the building blocks of proteins, and proteins have many important roles within and around cells. Bacteria regulate certain genes to ensure they have the right balance of different amino acids to survive. By controlling the availability of certain proteins that help them to make or collect certain amino acids, bacteria can control their overall amino acid balance.

Before a protein is made, a molecular machine called RNA polymerase must first copy the information in a gene to make a molecule called a messenger RNA (mRNA). The mRNA is then translated to make the protein from individual amino acids. In this process, each amino acid needs to be first attached to another molecule called a transfer RNA (tRNA). In many bacteria species, the mRNAs involved in making or transporting amino acids contain structures called T-boxes. These structures guide the RNA polymerase to make more of the mRNAs when the levels of the amino acid become too low. A T-box, however, does not sense the level of the amino acid directly. Instead it senses the number of tRNA molecules that do not carry an amino acid.

Zhang, Chetnani et al. examined a particular T-box interacting with tRNA using pairs of fluorescent dyes to detect distances between molecules. The T-box first recognizes a part of the tRNA called the anticodon to make sure it binds the correct type of tRNA. It then changes its shape to detect whether the tRNA is attached to an amino acid. This two-step process is driven by multiple structural elements within the T-box, and the flexibility of the T-box plays a critical role.

A cell's survival depends on it keeping amino acid levels under control. Understanding how bacteria do this could lead to new antibiotic drugs that target the T-box to kill cells. This study also provides insights into the workings of mRNA components like T-boxes – a type of riboswitch – which is an unusual means of controlling gene activity.

DOI: https://doi.org/10.7554/eLife.39518.002

riboswitches, binding of a charged tRNA to the T-box leads to rho independent transcription termination whereas an uncharged tRNA stabilizes the anti-terminator conformation and leads to transcription read-through (*Henkin, 2014*; *Zhang and Ferré-D'Amaré, 2015*). Whereas Stem I and the anti-terminator domain are highly conserved among T-box riboswitches, the region connecting them can vary. The *Bacillus subtilis glyQS* T-box riboswitch, involved in glycine regulation, represents one of the simplest T-box riboswitches (*Grundy et al., 2002b*) in which only a short linker and a small stem, Stem III, connect Stem I and the anti-terminator domain (*Figure 1*).

Recognition of tRNA by a T-box riboswitch involves three main structural elements of the tRNA: the anticodon region, the 'elbow' region formed by the conserved T- and D-loops, and the 3' NCCA sequence (*Figure 1*). The anticodon and elbow regions of the tRNA interact with Stem I directly. Stem I contains several phylogenetically conserved structural motifs (*Rollins et al., 1997*), including a K-turn motif, a specifier loop, a distal bulge, and an apical loop (*Rollins et al., 1997*) (*Figure 1*). Bioinformatics and structural analyses have collectively revealed the interactions between Stem I and the tRNA (*Grigg et al., 2013*; *Lehmann et al., 2013*; *Zhang and Ferré-D'Amaré, 2013*). Specifically, the co-crystal structures of Stem I/tRNA complexes show that Stem I flexes to follow closely the tRNA anticodon stem and interacts directly with the anticodon loop and the elbow through its proximal and distal ends, respectively (*Zhang and Ferré-D'Amaré, 2013*). The distal bulge and the apical loop fold into a compact structural module of interdigitated T-loops (*Chan et al., 2013*; *Krasilnikov and Mondragón, 2003*), which interact directly with conserved unstacked nucleobases at the tRNA elbow (*Grigg et al., 2013*; *Zhang and Ferré-D'Amaré, 2013*). In addition, the structures revealed that Stem I turns sharply around two hinge regions using a conserved dinucleotide bulge and the K-turn motif (*Grigg and Ke, 2013*; *Zhang and Ferré-D'Amaré, 2013*). Sensing of the aminoacylation state involves direct binding of the tRNA 3' end to a highly conserved bulge in the T-box, the t-box sequence (*Grundy et al., 2002a*) (*Figure 1*). A free NCCA end can base pair with the t-box sequence, enabling the anti-terminator conformation, whereas a charged NCCA end prevents the formation of the NCCA/t-box interactions, leading to the more stable terminator conformation (*Henkin, 2014*; *Zhang and Ferré-D'Amaré, 2015*). Importantly, discrimination between the charged and uncharged tRNA does not require any additional proteins, such as EF-Tu

(*Suddala et al., 2018*; *Zhang and Ferré-D'Amaré, 2014*), and is driven solely by RNA/RNA interactions.

While Small Angle X-ray Scattering (SAXS)-derived models of the entire *B. subtilis glyQS* T-box riboswitch in complex with tRNA are available (*Chetnani and Mondragón, 2017*; *Fang et al., 2017*), atomic-level structural details on the interactions between tRNA and the anti-terminator region are still lacking. In addition, there is a dearth of information on the kinetics of the binding process. Whereas it is clear that tRNA recognition involves several specific interactions, their binding temporal sequence remains elusive. In addition, it is unclear whether sensing of the 3' end of the tRNA involves any additional conformational changes in the T-box. Here, by introducing donor-acceptor fluorophore pairs at several locations in the tRNA and the T-box riboswitch, and using single-molecule fluorescence resonance energy transfer (smFRET), we demonstrate the temporal order of events in the trajectory of tRNA binding. Our results demonstrate that tRNA binds to the riboswitch in two steps, with its anticodon being recognized first, followed by NCCA binding accompanied with an inward motion of the 3' region of the T-box riboswitch, including Stem III and the anti-terminator stem, relative to Stem I. In addition, by introducing mutations at different locations of the T-box, we further show that the two-step binding kinetics is regulated by the modular structural elements in the T-box riboswitch.

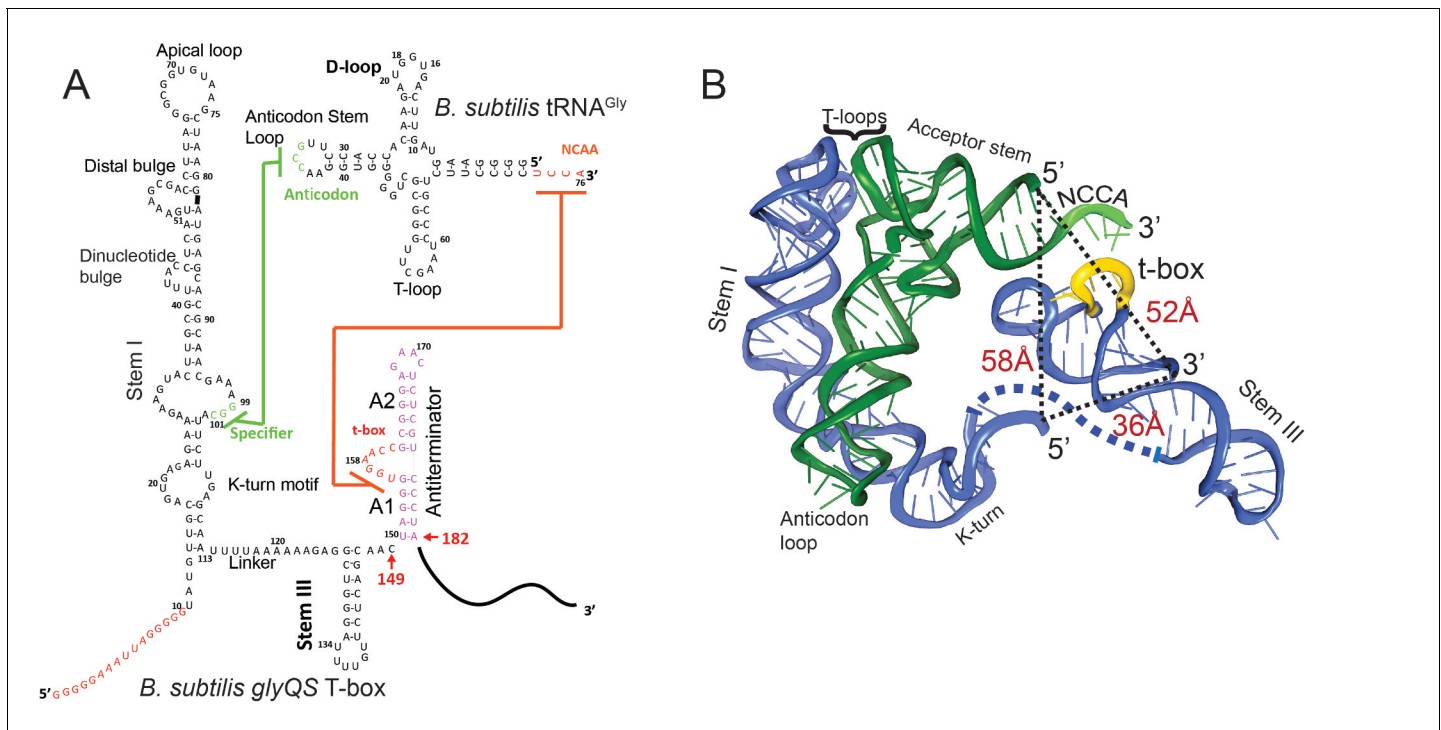

**Figure 1.** Secondary and tertiary structures of *B. subtilis glyQS* T-box riboswitch and tRNA[Gly]. (**A**) Secondary structure diagrams of the *B. subtilis glyQS* T-box riboswitch and *B. subtilis* tRNA[Gly] used in this study. Green and orange lines indicate interactions between the T-box specifier loop and the tRNA anticodon and between the T-box t-box sequence and the tRNA 3' NCCA, respectively. For the *glyQS* T-box sequence, the nucleotides in red were added for surface immobilization. (**B**) Ribbon diagram of a model of a complex between the *B. subtilis glyQS* T-box riboswitch (blue) and *B. subtilis* tRNA[Gly] (green) based on SAXS data (*Chetnani and Mondragón, 2017*). Distances between the 5' and 3' ends of the T-box and the 5' end of the tRNA[Gly] are shown (black dash lines). The NCCA sequence at the 3' end of the tRNA is shown in light green and the t-box sequence in the T-box is shown in yellow.

DOI: https://doi.org/10.7554/eLife.39518.003

The following figure supplements are available for figure 1:

**Figure supplement 1.** Native gel electrophoresis analysis of the folding of T-box constructs and binding of tRNA.

DOI: https://doi.org/10.7554/eLife.39518.004

**Figure supplement 2.** Isothermal Titration Calorimetry (ITC) of tRNA binding to T-box$_{182}$.

DOI: https://doi.org/10.7554/eLife.39518.005

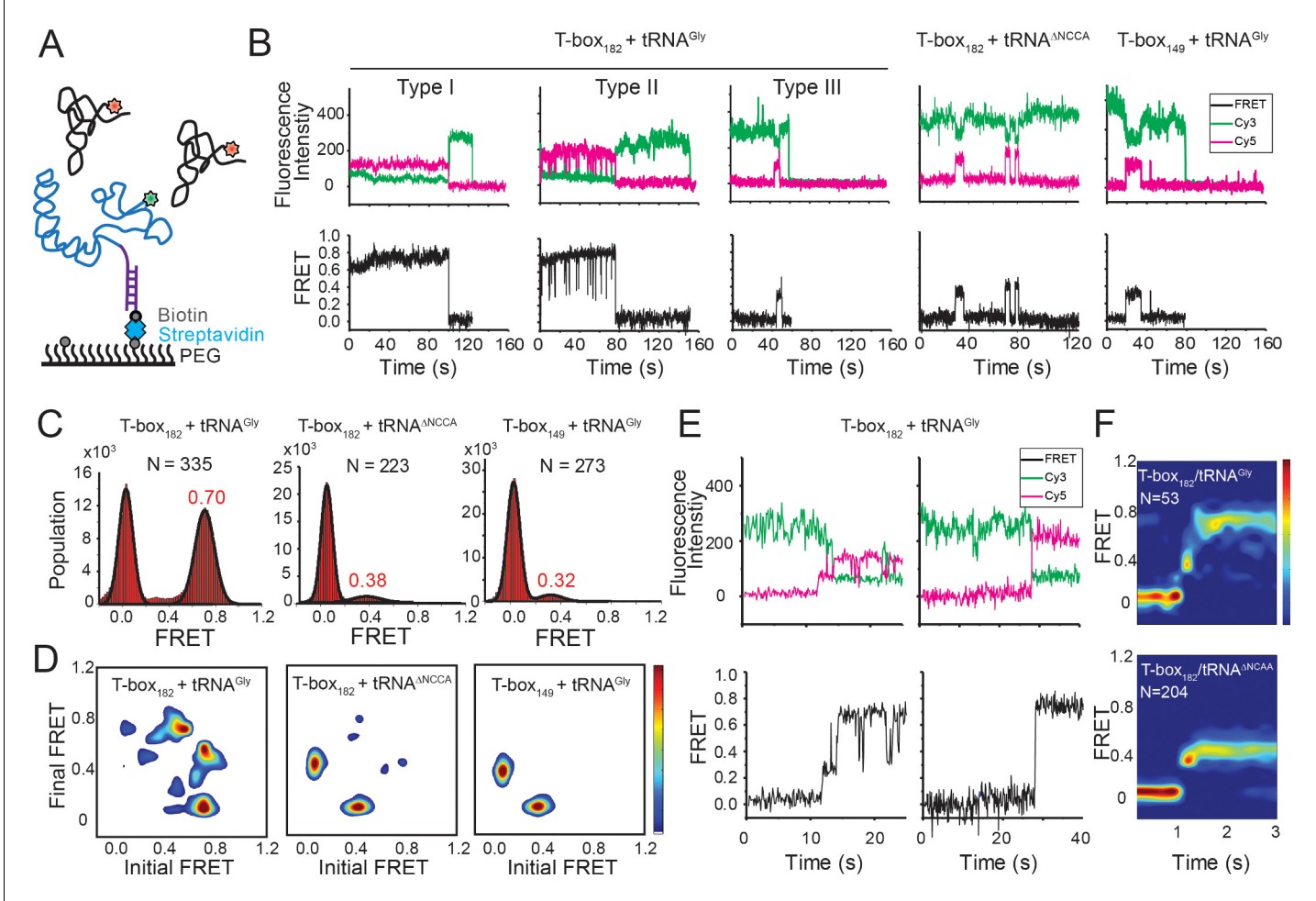

**Figure 2.** Two-step binding of uncharged tRNA to the *glyQS* T-box riboswitch. (**A**) FRET labeling scheme for the T-box and tRNA. Cy3 (green star) and Cy5 (red star) fluorophores are attached at the 3' of the T-box (blue) and the 5' of the tRNA (black), respectively. *glyQS* T-box riboswitch molecules are anchored on slides through a biotinylated DNA probe (purple) hybridized to a 5' extension sequence on the T-box. (**B**) smFRET *vs.* time trajectories of T-box$_{182}$-Cy3(3') with tRNA$^{Gly}$-Cy5, T-box$_{182}$-Cy3(3') with tRNA$^{\Delta NCCA}$-Cy5 and T-box$_{149}$-Cy3(3') with tRNA$^{Gly}$-Cy5. Cy3 and Cy5 fluorescence intensity traces (upper panel), and their corresponding smFRET traces calculated as $I_{Cy5} / (I_{Cy3}+I_{Cy5})$ (lower panel). (**C**) One-dimensional FRET histograms. FRET peaks are fit with a Gaussian distribution (black curve) and the peak centers are shown in red. 'N' denotes the total number of traces in each histogram from three independent experiments. (**D**) Transition density plot (TDP). Contours are plotted from white (less than 15% of the maximum population) to red (more than 85% of the maximum population). TDPs are generated from all smFRET traces from three independent experiments. (**E**) Representative smFRET trajectories showing real-time binding of tRNA$^{Gly}$-Cy5 to T-box$_{182}$-Cy3(3') in a steady-state measurement. Traces showing transitions from the unbound state (0 FRET) to fully bound state (0.7 FRET) through the partially bound state (0.4 FRET) (left) and unbound state directly to fully bound state (right). (**F**) Surface contour plot of time-evolved FRET histogram of T-box$_{182}$-Cy3(3') with tRNA$^{Gly}$-Cy5 (top) and tRNA$^{\Delta NCCA}$-Cy5 (bottom). Contours are plotted from blue (less than 5% of the maximum population) to red (more than 75% of the maximum population). 'N' denotes the total number of traces in each histogram from three independent experiments, which are a subset of traces showing real-time binding events in the steady-state measurements. Total numbers of traces in steady-state measurements are indicated in (**C**). Traces that reach the 0.7 FRET state (cutoff >0.55) are included in the plot for tRNA$^{Gly}$-Cy5 to reveal better the transition from the 0.4 to the 0.7 FRET state. Time-evolved FRET histograms of all traces are shown in *Figure 2—figure supplement 4D* for comparison.

DOI: https://doi.org/10.7554/eLife.39518.006

The following source data and figure supplements are available for figure 2:

**Source data 1.** Cy3 and Cy5 intensity traces from three repeats for T-box$_{182}$-Cy3(3') and tRNA$^{Gly}$-Cy5.
DOI: https://doi.org/10.7554/eLife.39518.014
**Source data 2.** Cy3 and Cy5 intensity traces from three repeats for T-box$_{182}$-Cy3(3') and tRNA$^{\Delta NCCA}$-Cy5.
DOI: https://doi.org/10.7554/eLife.39518.015
**Source data 3.** Cy3 and Cy5 intensity traces from three repeats for T-box$_{149}$-Cy3(3') and tRNA$^{Gly}$-Cy5.
DOI: https://doi.org/10.7554/eLife.39518.016
**Figure supplement 1.** Representative images of smFRET data for T-box-Cy3(3') and tRNA-Cy5 binding.
*Figure 2 continued on next page*

*Figure 2 continued*

DOI: https://doi.org/10.7554/eLife.39518.007

**Figure supplement 2.** Lifetime analyses of *glyQS* T-box-Cy3(3′) and tRNA-Cy5 interaction.

DOI: https://doi.org/10.7554/eLife.39518.008

**Figure supplement 3.** Intra-T-box FRET of T-box$_{182}$ and T-box$_{149}$.

DOI: https://doi.org/10.7554/eLife.39518.009

**Figure supplement 3—source data 1.** Cy3 and Cy5 intensity traces from two repeats for intra-T-box$_{182}$ FRET in the absence of tRNA.

DOI: https://doi.org/10.7554/eLife.39518.010

**Figure supplement 3—source data 2.** Cy3 and Cy5 intensity traces from two repeats for intra-T-box$_{149}$ FRET in the absence of tRNA.

DOI: https://doi.org/10.7554/eLife.39518.011

**Figure supplement 4.** Real-time flow experiment of the T-box$_{182}$-Cy3(3′) and tRNA$^{Gly}$-Cy5.

DOI: https://doi.org/10.7554/eLife.39518.012

**Figure supplement 4—source data 1.** Cy3 and Cy5 intensity traces from two repeats for T-box$_{182}$-Cy3(3′) and tRNA$^{Gly}$-Cy5 in the flow experiments.

DOI: https://doi.org/10.7554/eLife.39518.013

## Results

### Binding of cognate tRNA by the *glyQS* T-box results in two distinct FRET states

To observe directly the binding of tRNA to the T-box, we placed the donor dye (Cy3) on the 3′ end of a T-box fragment (T-box$_{182}$), and the acceptor dye (Cy5) on the 5′ end of the tRNA$^{Gly}$, where the subscript '182' denotes the length of the T-box construct (*Figure 1A*). In vitro transcribed and labeled T-box and tRNA were purified and refolded according to published procedures (*Chetnani and Mondragón, 2017*; *Zhang and Ferré-D'Amaré, 2013*) (*Figure 1—figure supplement 1*). Labeling of the tRNA at the 5′ end had a modest effect on the binding affinity (*Figure 1—figure supplement 2*). T-box$_{182}$ spans Stem I, the linker sequence, Stem III and the anti-terminator, but does not contain the terminator sequence, thereby preventing the transition to the terminator conformation. A short RNA extension sequence was added to the 5′ end of the T-box for surface immobilization (*Figure 2A*, *Supplementary file 1*). Single-molecule fluorescence images were recorded under equilibrium condition in the presence of 30 nM tRNA$^{Gly}$-Cy5. Binding of tRNA$^{Gly}$-Cy5 results in a major distribution of FRET values around 0.7, with $79 \pm 4\%$ of the traces showing a stable signal at 0.7 and $9 \pm 5\%$ traces sampling from 0.7 to 0.4 (*Figure 2B,C*). The SAXS model (*Chetnani and Mondragón, 2017*) predicts a distance between the labeling positions at the 3′ end of the T-box$_{182}$ and the 5′ end of the tRNA to be around 52 Å. (*Figure 1B*). Based on a Förster distance of 54–60 Å (*Ha et al., 2002*; *Hohng et al., 2004*), our measured FRET value is within the range of estimated FRET values (0.56–0.70). Therefore, we assign the 0.7 FRET state to be the fully bound state of the tRNA$^{Gly}$ by the T-box.

In order to assign the 0.4 FRET value to specific tRNA binding states, tRNA$^{Tyr}$-Cy5 and tRNA$^{\Delta NCCA}$-Cy5 ('$\Delta$NCCA' denotes a tRNA$^{Gly}$ with a deleted 3′ NCCA sequence) were flowed in the flow-chamber with pre-immobilized T-box$_{182}$-Cy3(3′) (3′ denotes that the label was added at the 3′ end). We did not observe any binding of tRNA$^{Tyr}$-Cy5 (*Figure 2—figure supplement 1*), confirming that recognition of the anticodon by the specifier region is required for tRNA binding. In the presence of tRNA$^{\Delta NCCA}$-Cy5, we observed a fluctuating signal between 0.4 and 0 FRET (*Figure 2B,D*), with a mean lifetime of the 0.4 FRET state of $3.6 \pm 0.6$ s and a mean waiting time before binding of $31.3 \pm 5.3$ s (*Figure 2—figure supplement 2B*). Taken together with the results from the tRNA$^{Gly}$, tRNA$^{\Delta NCCA}$ and tRNA$^{Tyr}$ binding experiments, we assign the 0.4 FRET state to a partially bound state where only the anticodon interactions have been established.

To further confirm the assignment of the FRET states, we generated T-box$_{149}$, where the anti-terminator sequence is truncated (*Figure 1A*, *Supplementary file 1*). Based on the structure model from the SAXS data (*Chetnani and Mondragón, 2017*) we predicted that a Cy3 dye placed either at the end of Stem III (T-box$_{149}$) or at the end of the anti-terminator stem (T-box$_{182}$) are localized in close proximity in three dimensions, further confirmed by the distance measurement using smFRET (*Figure 2—figure supplement 3*). Therefore, we expect that if tRNA$^{Gly}$-Cy5 can reach the same fully bound state in T-box$_{149}$ as in T-box$_{182}$, a high FRET state centered at 0.7 would be observed. However, using T-box$_{149}$-Cy3(3′) in combination with tRNA$^{Gly}$-Cy5, we again observed transient binding

of tRNA$^{Gly}$ with a FRET value centered at ~0.4 with the same mean lifetime as observed with the T-box$_{182}$-Cy3(3') and tRNA$^{\Delta NCCA}$-Cy5 combination (*Figure 2B–D*, *Figure 2—figure supplement 2C*). Therefore these two complexes (T-box$_{182}$ + tRNA$^{\Delta NCCA}$ and T-box$_{149}$ + tRNA$^{Gly}$) represent the same binding state of the tRNA, that is the state where binding of the anticodon to the specifier region has been established, but is unstable without the further interactions between the NCCA and the t-box region.

Collectively, our results suggest a two-step binding model involving the separate establishment of the interactions with the anticodon and the NCCA. The fact that tRNA$^{Tyr}$, which has a mismatched anticodon, but contains an intact NCCA 3' end, does not show any binding activity suggests that interactions with the anticodon precede the interactions with the NCCA end of the tRNA and are necessary for the establishment of the NCCA contacts. Without the interaction between the NCCA and the t-box sequence the binding of tRNA$^{Gly}$ is not stable. From the binding kinetics of tRNA$^{\Delta NCCA}$, we estimated the association rate constant ($k_1$) and the disassociation rate constant ($k_{-1}$) for the first binding step to be $(5.0 \pm 1.6) \times 10^5$ M$^{-1}$s$^{-1}$ and $0.28 \pm 0.05$ s$^{-1}$, respectively (Figure 5; Figure 6E).

## The transition from anticodon recognition to NCCA binding is rapid for uncharged tRNA

We classified smFRET traces for T-box$_{182}$-Cy3(3') in complex with tRNA$^{Gly}$-Cy5 into three types (*Figure 2B*): (I) traces stably sampling the 0.7 state (79 ± 4% of total traces), (II) traces transiently transitioning from the 0.7 state to the 0.4 state (9 ± 5%), and (III) traces only sampling the 0.4 state without reaching 0.7 state (12 ± 5%). The low percentage of Type III traces indicates that once the anticodon is recognized, the commitment to the next binding step, NCCA interactions, is high. The majority of the traces showed that the tRNA$^{Gly}$ remained mostly in the fully bound state (Type I) until the fluorophore photobleached, with the actual lifetime limited by photobleaching ($\tau_{0.7} > 24$ s, where $\tau_{0.7}$ denotes the lifetime of the 0.7 FRET state) (*Figure 2—figure supplement 2A*). The observation that tRNA$^{Gly}$ is able to transit from the fully bound state back to the partially bound state (Type II) suggests that the NCCA/t-box interaction can break occasionally (*Figure 2B*). We estimated the lifetime of the transiently sampled partially bound state to be $0.35 \pm 0.09$ s in the presence of full-length tRNA$^{Gly}$ (*Figure 2—figure supplement 2A*), ~10 fold shorter than the partially bound state in the presence of tRNA$^{\Delta NCCA}$.

While the majority of the T-box molecules were already bound to tRNA$^{Gly}$ before starting data acquisition, we could detect that some molecules show real-time binding during imaging acquisition. We observed only a few traces briefly sampling the 0.4 FRET state from the zero FRET (unbound) state before reaching the 0.7 FRET state, while most traces directly sampled the 0.7 FRET state without a detectable 0.4 FRET, likely due to our imaging time resolution (100 ms per frame). We post-synchronized the FRET traces at the transition point from the zero FRET state to the first sampled 0.4 FRET state, and plotted them in a time-evolved FRET histogram. From the time-evolved FRET histogram (*Figure 2F*), we estimated roughly that the upper limit of the lifetime spent at the 0.4 FRET state is ~100 ms, very rapidly followed by establishment of NCCA/t-box interactions. In contrast, tRNA$^{\Delta NCCA}$ could not pass the 0.4 FRET state. To capture better real-time binding, we performed a flow experiment, where tRNA$^{Gly}$-Cy5 was flowed into a chamber with immobilized T-box$_{182}$-Cy3(3') during imaging acquisition. The corresponding post-synchronized time-evolved FRET histogram again shows a fast transition into the fully bound state (*Figure 2—figure supplement 4*). In addition, the association rate constant of tRNA$^{Gly}$ in the real-time flow experiment is $(7.5 \pm 0.7) \times 10^5$ M$^{-1}$s$^{-1}$, consistent with the $k_1$ of tRNA$^{\Delta NCCA}$ and confirming that the NCCA end of the tRNA does not participate in the first binding step.

From the real-time binding kinetics of tRNA$^{Gly}$ to T-box$_{182}$, we estimated a transition rate constant from the partially bound state to the fully bound state ($k_2$) to be on the order of 10 s$^{-1}$ (*Figure 2F*, *Figure 2—figure supplement 4*). On the other hand, as transitions back to the partially bound state from the fully bound state were only observed in ~10% traces, we interpreted this to mean that the reverse transition rate constant ($k_{-2}$) is very small, and the second binding step in the wild-type (WT) T-box with uncharged tRNA$^{Gly}$ is close to irreversible (Figure 5; Figure 6E and see Discussion).

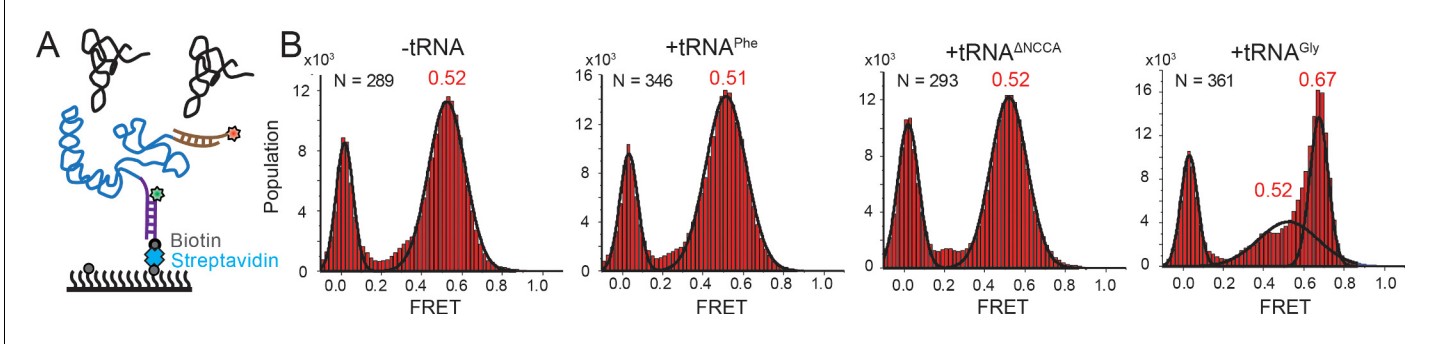

**Figure 3.** Conformational changes following tRNA binding in the *glyQS* T-box riboswitch revealed by an intra-T-box FRET pair. (A) Intra-T-box FRET scheme. Cy3 (green star) and Cy5 (red star) are attached at the 5' and 3' extensions of T-box (blue), respectively. (B) One-dimensional FRET histograms of T-box$_{182}$ alone, with tRNA$^{Phe}$, with tRNA$^{\Delta NCCA}$, and with tRNA$^{Gly}$. 'N' denotes the total number of traces in each histogram from three independent experiments.

DOI: https://doi.org/10.7554/eLife.39518.017

The following source data and figure supplements are available for figure 3:

**Source data 1.** Cy3 and Cy5 intensity traces from three repeats for intra T-box$_{182}$ FRET with 3' extension without tRNA.
DOI: https://doi.org/10.7554/eLife.39518.020
**Source data 2.** Cy3 and Cy5 intensity traces from three repeats for intra T-box$_{182}$ FRET with 3' extension in the presence of tRNA$^{Phe}$.
DOI: https://doi.org/10.7554/eLife.39518.021
**Source data 3.** Cy3 and Cy5 intensity traces from three repeats for intra T-box$_{182}$ FRET with 3' extension in the presence of tRNA$^{\Delta NCCA}$.
DOI: https://doi.org/10.7554/eLife.39518.022
**Source data 4.** Cy3 and Cy5 intensity traces from three repeats for intra T-box$_{182}$ FRET with 3' extension in the presence of tRNA$^{Gly}$.
DOI: https://doi.org/10.7554/eLife.39518.023
**Figure supplement 1.** Intra-T-box FRET of T-box$_{182}$ in response to tRNA binding.
DOI: https://doi.org/10.7554/eLife.39518.018
**Figure supplement 1—source data 1.** Cy3 and Cy5 intensity traces for intra-T-box$_{182}$ FRET in the presence of tRNA$^{Gly}$.
DOI: https://doi.org/10.7554/eLife.39518.019

## Establishment of the NCCA/t-box interaction is accompanied by conformational changes in the T-box riboswitch

We next investigated whether tRNA binding requires any conformational changes in the T-box itself. Using doubly labeled T-box$_{182}$, with Cy3 at the 3' end and Cy5 at the 5' hybridization extension, we observed a high FRET state (centered at ~0.75) in the absence of tRNA (*Figure 3—figure supplement 1*). Based on the structural model (*Chetnani and Mondragón, 2017*), we estimated the distance between the 5' and 3' ends of the T-box$_{182}$ to be ~36 Å (*Figure 1B*). Our measured FRET value is slightly less than the predicted FRET value (~0.90), likely due to the engineered 5' extension sequence used to immobilize the T-box. No noticeable change was detected upon incubation with unlabeled tRNA$^{Gly}$ (*Figure 3—figure supplement 1*), indicating that the 3' portion (Stem III plus the anti-terminator stem) does not move away from the 5' portion (Stem I). Given that the measured FRET efficiency of 0.75 is already located beyond the FRET sensitive region, it is unlikely that any inward motion of the 3' portion relative to the 5' could be detected. To overcome this limitation, we added extensions at both the 3' and 5' ends (*Figure 3A*, *Supplementary file 1*). ITC experiments suggest that addition of a 5' and/or a 3' extension sequences to the T-box does not affect tRNA binding (*Figure 1—figure supplement 2*). With this intra-T-box FRET scheme, we observed a FRET shift from ~0.5 to~0.65 when tRNA$^{Gly}$ was added (*Figure 3B*), indicating that the 3' half of the T-box moves closer to the 5' half, potentially with the T-box becoming more compact due to the presence of the cognate tRNA$^{Gly}$. Adding non-cognate tRNA$^{Phe}$ or tRNA$^{\Delta NCCA}$ gave similar FRET values as the T-box alone (*Figure 3B*), suggesting that the conformational change is associated with binding of both the anticodon and NCCA, not with anticodon recognition alone (Figure 5).

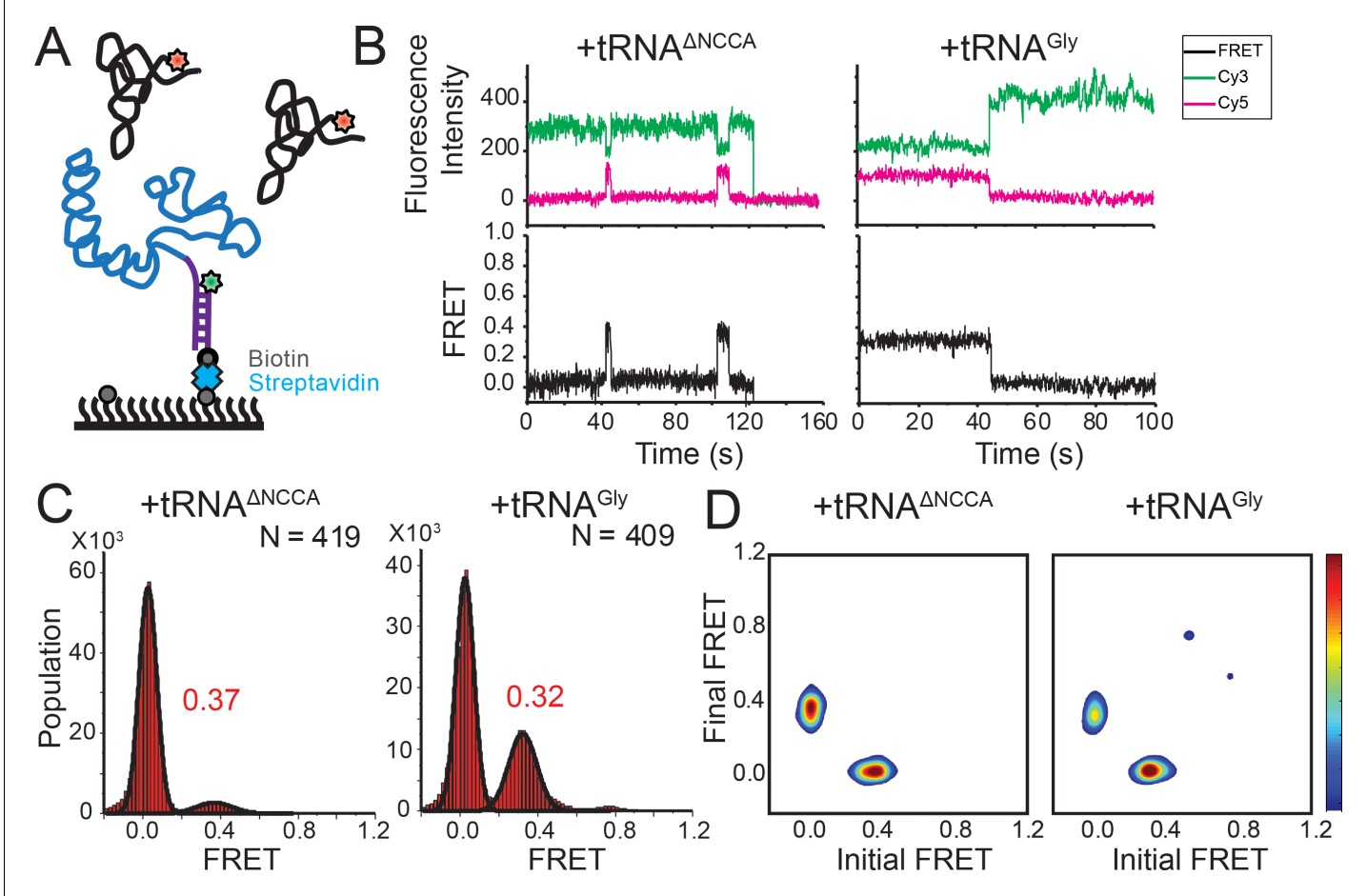

**Figure 4.** FRET between fluorophores at the 5' end of the *glyQS* T-box riboswitch and 5' end of tRNA$^{Gly}$ is insensitive to the two binding states. **(A)** Cy3 (green star) and Cy5 (red star) are attached at the 5' extension of the T-box (blue) and the 5' of the tRNA (black), respectively. **(B)** smFRET trajectories of T-box-Cy3(5') with tRNA$^{ΔNCCA}$-Cy5 (left) and tRNA$^{Gly}$-Cy5 (right). 'N' denotes the total number of traces in each histogram from three independent experiments. **(C)** One-dimensional FRET histograms of T-box-Cy3(5') with tRNA$^{ΔNCCA}$-Cy5 (left) and tRNA$^{Gly}$-Cy5 (right). **(D)** TDP of T-box-Cy3(5') with tRNA$^{ΔNCCA}$-Cy5 (left) and tRNA$^{Gly}$-Cy5 (right). Contours are plotted in the same way as in *Figure 2D*. TDPs are generated from all smFRET traces from three independent experiments.

DOI: https://doi.org/10.7554/eLife.39518.024

The following source data and figure supplement are available for figure 4:

**Source data 1.** Cy3 and Cy5 intensity traces from three repeats for T-box$_{182}$-Cy3(5') with tRNA$^{Gly}$-Cy5.

DOI: https://doi.org/10.7554/eLife.39518.026

**Source data 2.** Cy3 and Cy5 intensity traces from three repeats for T-box$_{182}$-Cy3(5') with tRNA$^{ΔNCCA}$-Cy5.

DOI: https://doi.org/10.7554/eLife.39518.027

**Figure supplement 1.** Lifetime analyses of *glyQS* T-box-Cy3(5') and tRNA-Cy5 interaction.

DOI: https://doi.org/10.7554/eLife.39518.025

## The NCCA end of the uncharged tRNA maintains its relative position to the K-turn region during the second binding step

Using the above two FRET pairs, we observed that the 3' portion of the T-box moves towards the base of Stem I as well as the NCCA end of the tRNA during the second binding step. To ascertain whether the NCCA end of the tRNA also moves relative to the base of Stem I, we measured FRET between a Cy3 placed at the 5' end of the T-box (T-box$_{182}$-Cy3(5')) and tRNA$^{Gly}$-Cy5 (*Figure 4A*). Using this FRET pair, binding of both tRNA$^{Gly}$ and tRNA$^{ΔNCCA}$ generated a similar FRET value centered at ~0.35 (*Figure 4B,C*). However, the FRET traces behaved differently for these two tRNA molecules. For tRNA$^{ΔNCCA}$, the signal fluctuated between zero and 0.35 (*Figure 4D*), with a lifetime of the 0.35 FRET state of 4.5 ± 1.0 s, reminiscent of the 0.4 FRET state using the tRNA/T-box$_{182}$-Cy3(3')

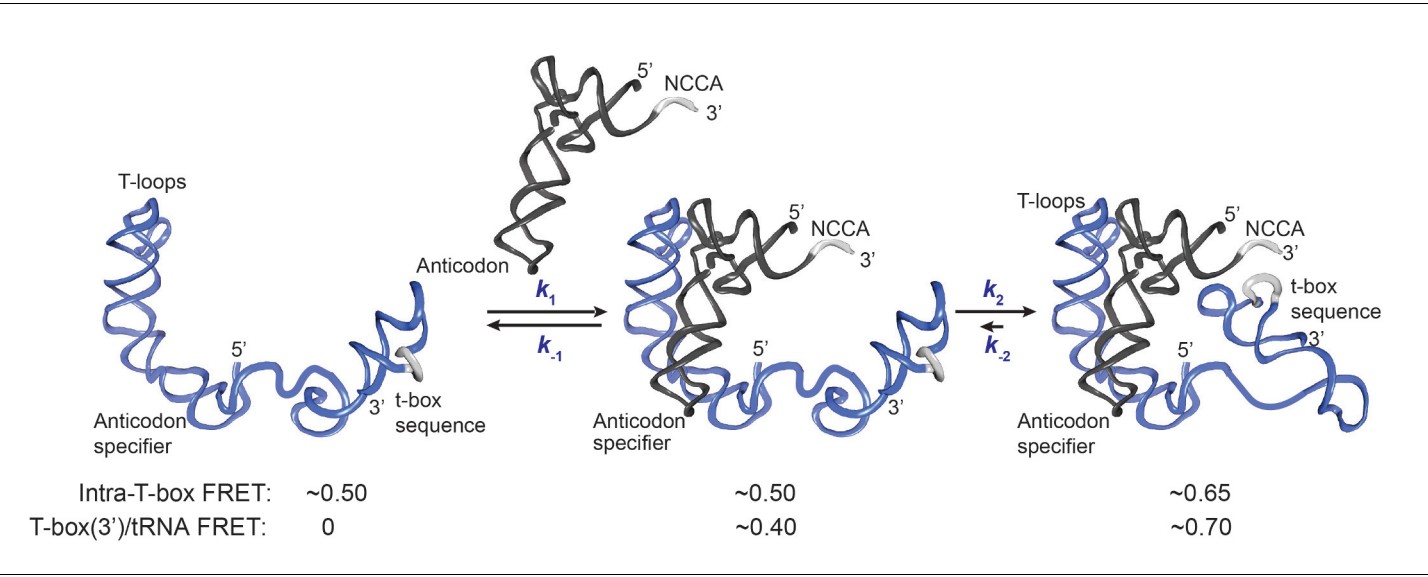

**Figure 5.** Kinetic model for the two-step binding of *glyQS* T-box riboswitch and uncharged tRNA<sup>Gly</sup>. Details of the model are described in the text. Rate constants are summarized in *Figure 6E*.

DOI: https://doi.org/10.7554/eLife.39518.028

FRET pair (*Figure 4—figure supplement 1*). For tRNA$^{Gly}$, the signal was more stably centered at 0.35 (*Figure 4B*). Since the tRNA-Cy5/T-box$_{182}$–Cy3(5') FRET pair cannot distinguish the partially bound from the fully bound state, we fit the lifetime with a double-exponential decay. The fast dissociation fraction has a lifetime of 3.9 ± 0.7 s (46 ± 21% of population), consistent with the lifetime for the partially bound state, and the low dissociation fraction has a lifetime of 15.7 ± 0.8 s (54 ± 21%), representing the stable fully bound state (*Figure 4—figure supplement 1*). Overall, the measurements with the tRNA-Cy5/T-box–Cy3(5') FRET pair further validate the two-step binding model and reveal that the NCCA end of the uncharged tRNA maintains its relative position to the base of Stem I during the second binding step.

## A mutation in the T-loop region affects the first binding step but has minimal effect on the second binding step

The interdigitated T-loops structure formed by the interactions between the distal bulge and the apical loop at the distal end of Stem I has been shown to be important for tRNA binding (*Grigg et al., 2013*; *Lehmann et al., 2013*; *Zhang and Ferré-D'Amaré, 2013*). Specifically, C56 of the T-box stacks on a nucleobase in the D-loop of the tRNA, and a point mutation of C56 to U has been shown to reduce the tRNA binding affinity by ~40 fold (*Zhang and Ferré-D'Amaré, 2013*). We introduced the same mutation in the T-box$_{182}$ backbone (T-box$_{C56U}$) (*Figure 6A*, *Supplementary file 1*). The smFRET trajectories for tRNA$^{Gly}$ binding to T-box$_{C56U}$ are overall similar to the trajectories for WT T-box$_{182}$, with a majority of traces (73 ± 6% of total traces) showing stable binding at 0.7 FRET state, and 13 ± 4% of the traces showing transitions back to the 0.4 FRET state (*Figure 6C,D*). $\tau_{0.7}$ was estimated to be at least ~23 s (limited by the photobleaching of the fluorophore) (*Figure 6E*). Post-synchronized time-evolved histogram on the subset of traces that demonstrated real-time binding shows fast transition to the fully bound state (*Figure 6—figure supplement 1*). Comparison of tRNA$^{Gly}$ binding to T-box$_{C56U}$ and T-box$_{182}$ suggest that the C56U mutation does not affect the second binding step. To investigate whether the mutation at the T-loop region affects the first binding step, we analyzed the binding and dissociation of tRNA$^{\Delta NCCA}$-Cy5 to T-box$_{C56U}$-Cy3(3'). We found that the $k_1$ of tRNA binding to T-box$_{C56U}$ was roughly 20-fold slower compared to tRNA binding to T-box$_{182}$, and the dissociation was roughly 2.5-fold faster compared to T-box$_{182}$ (*Figure 6E*), leading to a ~ 50 fold higher dissociation constant for the first binding step. Our results suggest that the T-loop region of the T-box is critical during the first binding

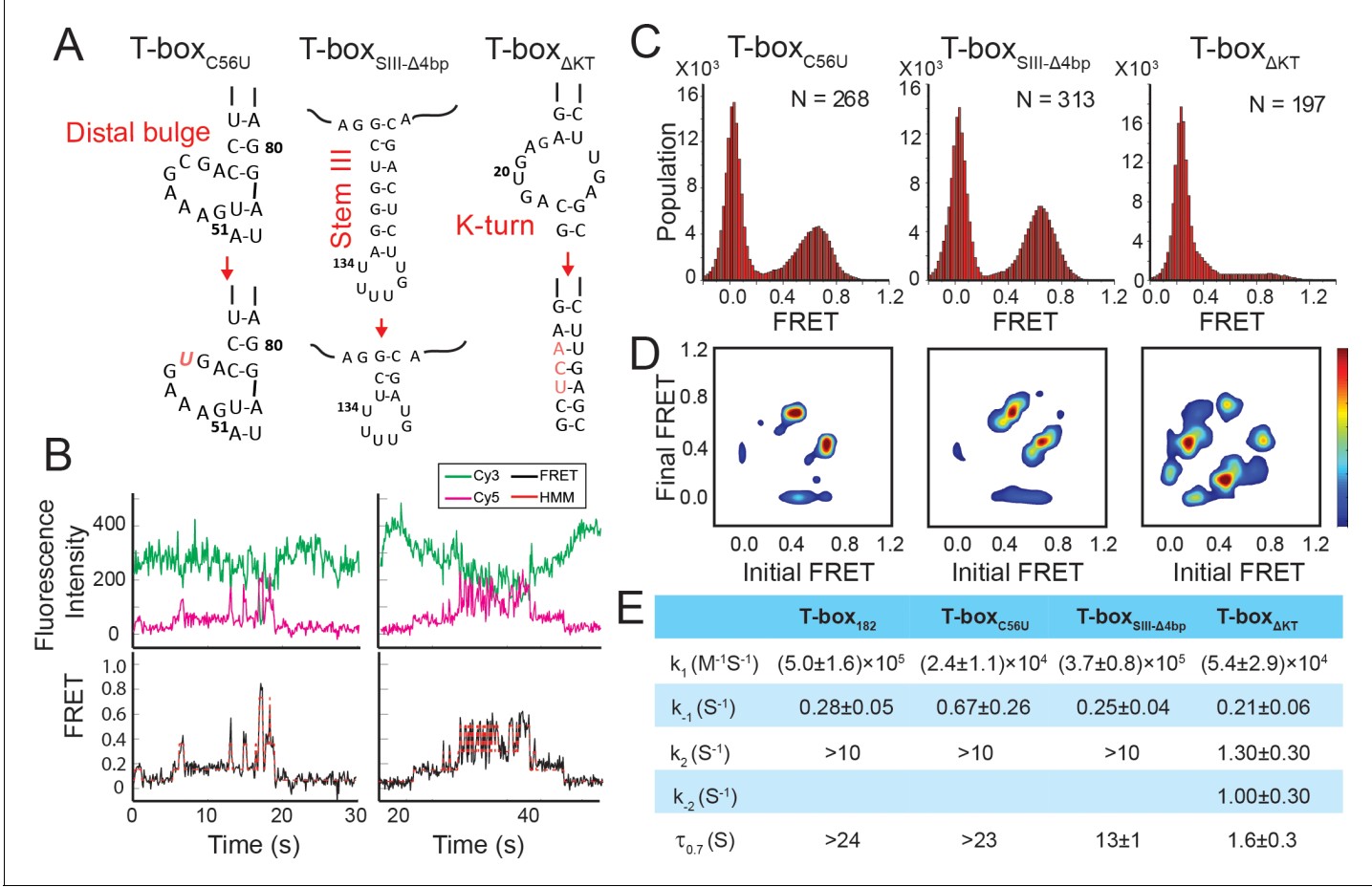

**Figure 6.** Regulation of the tRNA^Gly binding kinetics by structural elements in the *glyQS* T-box riboswitch. (A) Schematic diagram of three different mutations introduced to the T-box_182 backbone (T-box_C56U, T-box_SIII-Δ4bp and T-box_ΔKT). (B) Representative smFRET traces of T-box_ΔKT-Cy3(3') and tRNA^Gly-Cy5. (C) FRET histograms of the T-box mutants with tRNA^Gly-Cy5. 'N' denotes the total number of traces in each histogram from three independent experiments. (D) TDP of the T-box mutants with tRNA^Gly-Cy5. TDPs are generated from all smFRET traces from three independent experiments and are plotted in the same way as in *Figure 2D*. (E) Table of kinetic parameters for tRNA^Gly-Cy5 binding to different T-box constructs. $k_{-1}$, $k_2$, and $k_{-2}$ of T-box_ΔKT-Cy3(3') are apparent rate constants estimated to allow comparison as described in Materials and methods. All rate constants are reported as mean ± standard deviation (S.D.) from three or four independent experiments.

DOI: https://doi.org/10.7554/eLife.39518.029

The following source data and figure supplements are available for figure 6:

**Source data 1.** Cy3 and Cy5 intensity traces from three repeats for T-box_C56U-Cy3(3') with tRNA^Gly-Cy5.
DOI: https://doi.org/10.7554/eLife.39518.035
**Source data 2.** Cy3 and Cy5 intensity traces from four repeats for T-box_SIII-Δ4bp-Cy3(3') with tRNA^Gly-Cy5.
DOI: https://doi.org/10.7554/eLife.39518.036
**Source data 3.** Cy3 and Cy5 intensity traces from three repeats for T-box_ΔKT-Cy3(3') with tRNA^Gly-Cy5.
DOI: https://doi.org/10.7554/eLife.39518.037
**Source data 4.** Cy3 and Cy5 intensity traces from three repeats for T-box_C56U-Cy3(3') with tRNA^ΔNCCA-Cy5.
DOI: https://doi.org/10.7554/eLife.39518.038
**Source data 5.** Cy3 and Cy5 intensity traces from three repeats for T-box_SIII-Δ4bp-Cy3(3') with tRNA^ΔNCCA-Cy5.
DOI: https://doi.org/10.7554/eLife.39518.039
**Source data 6.** Cy3 and Cy5 intensity traces from three repeats for T-box_ΔKT-Cy3(3') with tRNA^ΔNCCA-Cy5.
DOI: https://doi.org/10.7554/eLife.39518.040
**Figure supplement 1.** tRNA-Cy5 binding to T-box_C56U-Cy3(3').
DOI: https://doi.org/10.7554/eLife.39518.030
**Figure supplement 2.** tRNA-Cy5 binding to T-box_SIII-Δ4bp-Cy3(3').
DOI: https://doi.org/10.7554/eLife.39518.031
**Figure supplement 3.** tRNA-Cy5 binding to T-box_ΔKT-Cy3(3').

*Figure 6 continued on next page*

*Figure 6 continued*

DOI: https://doi.org/10.7554/eLife.39518.032

**Figure supplement 4.** Normalized FRET trace percentage of T-box-Cy3(3') and tRNA$^{\Delta NCCA}$-Cy5.

DOI: https://doi.org/10.7554/eLife.39518.033

**Figure supplement 5.** Determination of $k_1$.

DOI: https://doi.org/10.7554/eLife.39518.034

step, potentially aiding in anticodon recognition, but does not contribute significantly to the second binding step.

## A truncation of stem III has a minor effect on tRNA binding

The functional role of Stem III is unclear. It has been speculated that Stem III might serve as a transcription stalling site to allow co-transcriptional folding and regulation of the T-box riboswitch (*Grundy and Henkin, 2004*; *Zhang and Landick, 2016*). In addition, a SAXS data-derived model suggested coaxial stacking of Stem III and the anti-terminator stem, leading to a plausible role of Stem III in stabilizing the anti-terminator conformation in the presence of uncharged tRNA$^{Gly}$ (*Chetnani and Mondragón, 2017*). To investigate the latter hypothesis, we generated a T-box mutant (T-box$_{SIII-\Delta 4bp}$), in which four base pairs in Stem III are deleted to significantly shorten its length (*Figure 6A*, *Supplementary file 1*). smFRET studies using T-box$_{SIII-\Delta 4bp}$-Cy3(3') with tRNA$^{\Delta NCCA}$-Cy5 and tRNA$^{Gly}$-Cy5 revealed insignificant difference in overall kinetics in the first and second step bindings (*Figure 6C–E*, *Figure 6—figure supplement 2*). Noticeably, the $\tau_{0.7}$ was around 50% shorter than that for the T-box$_{182}$ (*Figure 6E*), indicating that Stem III may contribute to the stabilization of the fully bound state, potentially through coaxial stacking with the anti-terminator stem, but the effect is minor.

## A K-turn mutation affects both binding steps

We next investigated the role of the K-turn in regulating tRNA binding kinetics. We disrupted the K-turn (T-box$_{\Delta KT}$) by changing the six bulged nucleotides to three nucleotides (UCA) to replace the K-turn with a three base pair stem (*Figure 6A*, *Supplementary file 1*). In contrast to binding of tRNA$^{Gly}$ to T-box$_{182}$, binding to T-box$_{\Delta KT}$ results in three FRET states centered on ~0.2, 0.4, and 0.7. (*Figure 6B*). While the exact boundary of each FRET state is difficult to determine accurately from the FRET histogram (*Figure 6C*), a transition density plot (TDP) clearly reveals interconversion between the 0.2, 0.4, and 0.7 states (*Figure 6D*), with transitions between the 0.2 and 0.4 FRET states, and between the 0.4 and 0.7 FRET states more populated. Binding of tRNA$^{\Delta NCCA}$-Cy5 to T-box$_{\Delta KT}$, on the other hand, leads to the loss of population of the 0.7 state; however, both the 0.2 and 0.4 FRET states and fluctuations between these two states are frequently sampled (*Figure 6—figure supplement 3A,B*).

Comparing the tRNA$^{Gly}$ and tRNA$^{\Delta NCCA}$ binding, we speculate that both the 0.2 and 0.4 FRET states observed in the case of the T-box$_{\Delta KT}$ represent the partially bound state in which only the anticodon and elbow are recognized. In contrast to T-box$_{182}$, T-box$_{\Delta KT}$, with the K-turn replaced by an extension of Stem I, could potentially favor a relaxed conformation of Stem I, as observed in the NMR structure of an isolated K-turn and specifier loop domain (*Wang and Nikonowicz, 2011*), generating a lower FRET value centered at 0.2. However, the sampling of the 0.4 FRET state in the T-box$_{\Delta KT}$ construct suggests that the interactions between the specifier and anticodon of the tRNA may transiently force open the extended base pair region and bend the T-box to adopt a similar conformation to the one observed in T-box$_{182}$. The lifetimes of the 0.4 state before transition back to the 0.2 state and before transition forward to the 0.7 state are $0.30 \pm 0.03$ s and $0.13 \pm 0.05$ s, respectively (*Figure 6—figure supplement 3C*), indicating that this forced bent state is energetically unfavorable. However, this 0.4 FRET state is very likely to be required for the NCCA/t-box interaction to occur, as in the presence of tRNA$^{Gly}$ the transitions from the 0.2 FRET to the 0.7 FRET state often pass through the 0.4 FRET state (*Figure 6B*). Furthermore, we observed a small region in the TDP corresponding to direct transitions between the 0.2 and 0.7 states. Given the very short lifetime of the 0.4 state, which is close to the time resolution of our experiments, the 0.2 to 0.7 state

transition is likely to represent populations whose 0.4 FRET state lifetime is even shorter than the time resolution of the experiment.

The 3-step kinetic scheme for tRNA binding to T-box$_{\Delta KT}$ is presented in *Figure 6—figure supplement 3C*. The association rate constant $k_1$ (($5.4 \pm 2.9$) x$10^4$ M$^{-1}$s$^{-1}$), estimated from binding of tRNA$^{\Delta NCCA}$ to the T-box$_{\Delta KT}$, is ~9 fold smaller than binding to T-box$_{182}$, suggesting that disruption of the K-turn affects anticodon recognition. Considering both the 0.2 and 0.4 FRET state as the partially bound state in T-box$_{\Delta KT}$, the apparent dissociation rate from the partially bound state $k_{-1\_app}$ of tRNA$^{\Delta NCCA}$ to T-box$_{\Delta KT}$ is $0.21 \pm 0.06$ s$^{-1}$, similar to that for T-box$_{182}$, suggesting that disruption of the K-turn does not affect the stability of the partially bound state. The 0.7 FRET state observed for T-box$_{\Delta KT}$ in the presence of tRNA$^{Gly}$ is consistent with the FRET value for the fully bound state in the WT T-box, indicating that the NCCA/t-box interactions in T-box$_{\Delta KT}$ can still be formed. However, tRNA$^{Gly}$ bound to T-box$_{\Delta KT}$ ($\tau_{0.7} = 1.6\pm0.3$ s) is at least 15-fold less stable compared to tRNA$^{Gly}$ bound to T-box$_{182}$ ($\tau_{0.7}$ >24 s). Furthermore, in contrast to the signal observed for tRNA$^{Gly}$ binding to T-box$_{182}$, in which fewer than 10% of the FRET traces show transitions back to 0.4 FRET, in T-box$_{\Delta KT}$ the vast majority of the traces show backward transitions to the 0.4 and 0.2 FRET states, contributing largely to the instability of the fully bound state. Based on the transition rates between 0.2, 0.4, and 0.7, we estimated the apparent forward ($k_{2\_app}$) and reverse ($k_{-2\_app}$) transition rates between the partially bound and the fully bound state of the T-box$_{\Delta KT}$ to be $1.3 \pm 0.3$ s$^{-1}$ and $1.0 \pm 0.3$ s$^{-1}$, respectively (*Figure 6E*, *Figure 6—figure supplement 3C*). The dramatically reduced $k_{2\_app}$ and increased $k_{-2\_app}$ in T-box$_{\Delta KT}$ leads to a ~ 150 fold change in the equilibrium constant in the second binding step compared to T-box$_{182}$, implying that inflexibility of the K-turn region largely inhibits the conformational change in the T-box required to form the NCCA/t-box interaction, and strongly destabilizes the fully bound state.

## Discussion

T-box riboswitches represent a unique class of riboswitches as they recognize a macromolecule and require interactions at multiple spatially separated sites on the ligand, unlike other riboswitches that respond to the binding of small ligands. Previous structural studies suggested that tRNA recognition by a T-box riboswitch is a bipartite process (*Grigg and Ke, 2013*; *Zhang and Ferré-D'Amaré, 2013*) with Stem I largely responsible for discriminating non-cognate tRNAs while the t-box sequence in the expression platform senses the charged state of the tRNA. We used smFRET to elucidate the binding kinetics of tRNA$^{Gly}$ by the *glyQS* T-box riboswitch. With three FRET pairs between different T-box riboswitch and tRNA ligand constructs, our data collectively reveal a two-step model of uncharged tRNA$^{Gly}$ binding to the *glyQS* T-box riboswitch (*Figure 5*). The first binding step involves recognition of the anticodon of the tRNA by the specifier sequence located in Stem I of the T-box riboswitch, leading to a partially bound state. In the second step, the 3' end of the T-box docks into the NCCA end of the tRNA through interactions with the t-box sequence, which leads to a fully bound state. Without the NCCA interaction, the binding of tRNA is unstable, with a mean lifetime of ~4 s, whereas with interactions both with the anticodon and the NCCA end, the binding of tRNA is very stable, with a mean lifetime >24 s. It should be noted that the latter lifetime is likely to be much longer for two reasons: first, the measurement is limited by fluorophore photobleaching, i.e. the loss of signal is more likely to be due to photobleaching rather than the actual tRNA dissociation; second, the Cy5 label placed at the 5' end of the tRNA reduces the overall binding affinity by ~4 fold (*Figure 1—figure supplement 2*), likely because the fluorophore impairs the NCCA/t-box interactions to some extent. In addition, an intra-T-box FRET pair at the 5' and 3' ends of the T-box demonstrates that, while the T-box is largely pre-organized in a folded state before tRNA binding, it still exhibits conformational rearrangement in a tRNA-dependent manner. Specifically, the 3' half of the T-box (including Stem III and the anti-terminator) moves inward relative to the 5' half (Stem I) of the T-box to accommodate the interaction with the NCCA end in the second binding step.

While our manuscript was in preparation, Suddala et al. (*Suddala et al., 2018*) reported a related single-molecule study on tRNA binding to the *glyQS* T-box riboswitch and proposed a similar two-step binding model. By using colocalized signals from the bound tRNA and the immobilized T-box, Suddala et al. (*Suddala et al., 2018*) uncovered two binding states distinguished by different dissociation rates of the tRNA, aided by using a Stem I-only mutant that cannot interact with the NCCA end of the tRNA. Specifically, in the model of Suddala et al. (*Suddala et al., 2018*) binding of the

anticodon of the uncharged tRNA results in a relatively unstable state (with a lifetime of ~4–5 s), consistent with the partially bound state in our model, while binding both the anticodon and the NCCA end of the tRNA results in a stable state, consistent with the fully bound state in our model. Although both studies propose consistent kinetic models, in the study by Suddala *et al.* (*Suddala et al., 2018*), the FRET pair dyes are attached at the variable loop of the tRNA and the 3' or 5' end of the *glyQS* T-box, and do not generate different FRET signals to distinguish the partially bound state from the fully bound state. Therefore neither transitions between the partially bound and the fully bound states, nor the order of events during tRNA binding can be resolved in the study (*Suddala et al., 2018*). In contrast, by employing a FRET pair located at the 5' end of the tRNA and the 3' end of the *glyQS* T-box, we observed directly two FRET states corresponding to the recognition of the anticodon (0.4 FRET) and the binding of the NCCA (0.7 FRET), therefore our study allows the discrimination between different states and generates a more complete kinetic framework describing the full trajectory of the tRNA binding process. Our data reveal that anticodon recognition precedes the NCCA end interactions, and that after anticodon recognition the commitment to further establishment of the NCCA/t-box interaction is high. The T-box/tRNA$^{Gly}$ complex transits rapidly from the partially bound state to the fully bound state, with a rate constant ($k_2$) on the order of 10 s$^{-1}$. Interestingly, our data also reveal that in the fully bound state the NCCA/t-box interaction is not highly-stable or ultra-stable. Brief disruption of the NCCA/t-box interactions can occur, but transition back to the fully bound state is rapid,~10 fold faster than dissociation of the tRNA from the partially bound state. Therefore tRNA can remain bound during the breaking and reforming of the NCCA/t-box interaction. However, such transient breaking of NCCA/t-box interaction was only observed in ~10% of the total population, suggesting that the reverse transition rate constant ($k_{-2}$) is overall very small.

Two-step binding mechanisms have been observed in a variety of protein or nucleic acid mediated biological process, including T-cell receptor (TCR) recognition of the major histocompatibility complex (MHC) presenting peptides, where TCRs scan the MHC scaffold first, followed by sensing of specific MHC-presenting peptides (*Wu et al., 2002*); interaction of the signal recognition particle (SRP) receptor with the membrane, where SRP receptors interact with the membrane in a dynamic mode followed by an SRP-induced conformational transition into a stable binding mode (*Hwang Fu et al., 2017*), DNA interrogation by CRISPR Cas9-crRNA, where Cas9-crRNA recognizes the protospacer adjacent motif (PAM) on the target DNA followed by sensing of the spacer sequence and triggering R-loop formation (*Sternberg et al., 2014*); and RNA-induced silencing complexes (RISCs) binding to their mRNA targets, where dynamic sampling of the 'sub-seed' region occurs before targeting stably across the full seed region (*Chandradoss et al., 2015*; *Herzog and Ameres, 2015*; *Salomon et al., 2015*; *Salomon et al., 2016*). In all of these cases, a two-step binding mechanism provides a good balance between sensitivity and specificity. T-box riboswitches fold and function co-transcriptionally in the cell, and hence we propose that a two-step kinetic model is kinetically beneficial during co-transcriptional folding of the T-box and sensing of the tRNA ligand. First, considering that the intracellular concentration of tRNA is on the order of µM in bacteria (*Avcilar-Kucukgoze et al., 2016*; *Dong et al., 1996*; *Emilsson and Kurland, 1990*), the binding of the tRNA ligand to the T-box riboswitch is rate limited by the first step. Pausing at Stem III in the presence of NusG in vitro was estimated to last 30–60 s (*Grundy and Henkin, 2004*). Assuming an in vivo concentration of tRNA ~1 µM, the apparent tRNA binding rate in the first step is ~0.5 s$^{-1}$, which is sufficient for the uncharged tRNA to reach the partially bound state during the pausing. Second, bacterial RNA polymerase (RNAP) elongates at a rate of 40–50 nt/s (*Mosteller and Yanofsky, 1970*; *Vogel and Jensen, 1994*), and the length between terminator and anti-terminator sequences is ~40 nts; therefore the decision between termination and anti-termination needs to be made within ~0.5–1 s. On the one hand, the relatively slow dissociation ($k_{-1}$) from the partially bound state can ensure the tRNA ligand stays bound until the completion of the anti-terminator sequence. On the other hand, very rapid transition into the fully bound state ($k_2$) helps secure the interaction between the NCCA end of the tRNA and the t-box sequence and trap the T-box in the anti-terminator conformation before completion of transcription of the terminator sequence, which is immediately downstream. Finally, the fully bound state has a lifetime of at least 24 s, allowing RNAP to elongate more than 1000 nts before the tRNA dissociates. Therefore, the second binding step in the WT T-box with uncharged tRNA$^{Gly}$ is close to irreversible in such biological setting. It is worth mentioning that Suddala *et al.* (*Suddala et al., 2018*) revealed that the uncharged T-box/tRNA$^{Gly}$ complex exists in two

populations: a stable complex, with a tRNA dissociation rate of ~0.03 s$^{-1}$, and an ultra-stable complex, with a tRNA dissociation rate <$2.4 \times 10^{-4}$ s$^{-1}$. Our estimation of the lifetime of the fully bound state was limited by the introduction of Cy5 to the 5' end of the tRNA and fluorophore photobleaching, and therefore our experiments cannot distinguish between the stable and ultra-stable complexes. Nevertheless, both potential configurations in the fully bound state are stable enough to allow transcription read-through.

Based on the known structures of Stem I in complex with tRNA$^{Gly}$ (*Grigg and Ke, 2013*; *Zhang and Ferré-D'Amaré, 2013*), it was hypothesized that the intra-T-box conformational change is likely to involve the K-turn region, which sits at the junction between the 5' and 3' portions of the T-box. Our observation that the NCCA end of the uncharged tRNA moves towards the anti-terminator stem (revealed by the T-box$_{182}$-Cy3(3') and tRNA$^{Gly}$-Cy5 FRET), but maintains its relative position to the K-turn region (revealed by the T-box$_{182}$-Cy3(5') and tRNA$^{Gly}$-Cy5 FRET) during the second binding step is consistent with the role of the K-turn region as the hinge of the intra-T-box conformational change. In addition, such conformational changes mediated by the K-turn region are Mg$^{2+}$ dependent, as suggested by Suddala *et al.* (*Suddala et al., 2018*). Importantly, our results hint at the critical role of the K-turn region in promoting the fast coordination between anticodon sensing and locking of the NCCA end. Crystal structures of the Stem I/tRNA complex (*Grigg and Ke, 2013*; *Zhang and Ferré-D'Amaré, 2013*) show that Stem I flexes around the K-turn region and that this flexing seems to be important to establish the interactions between the anticodon and specifier sequence. Using a mutant where the K-turn region is removed, our data show that in the absence of the K-turn motif both the first and second binding steps are affected, but with a much more dramatic effect on the second step. In this specific K-turn mutant, where the K-turn is replaced by an extended Stem I helix, the conformational change in the T-box that brings closer the 3' and 5' portions becomes highly energetically unfavorable, leading to a ~ 10 fold reduction in $k_2$, and ~15 fold destabilization of the fully bound state. Our kinetic measurement of the K-turn mutant explains the in vivo loss-of-function K-turn mutants (*Winkler et al., 2001*), and emphasizes the importance of the flexing around the K-turn region and the associated conformational changes in the T-box itself in the overall recognition and binding process.

Our results with various T-box mutants demonstrate that the T-box structural elements involved in tRNA recognition can drive the two-step binding process in a modular fashion. For example, in the first binding step, interactions with Stem I involve both anticodon recognition and the contacts between the interdigitated T-loops in Stem I and the elbow region (D- and T-loops) of the tRNA (*Grigg and Ke, 2013*; *Zhang and Ferré-D'Amaré, 2013*). When introducing a point mutation that impairs the T-loops/elbow region interaction (*Zhang and Ferré-D'Amaré, 2013*), we observed a dramatic decrease in the association rate constant and a moderate increase in the dissociation rate constant, leading to an overall ~50 fold reduction on the binding affinity for the first step, however the second step is unaffected. This observation suggests that establishment of the T-loops/elbow interactions is an important part of the Stem I/tRNA recognition process in the first binding step, but does not play any role in NCCA recognition in the second binding step. Similarly, truncation of Stem III has a minor effect on the stability of the fully bound state, but with no influence on the first binding step. Finally, the K-turn region (discussed above), which links the 5' and 3' portions of the T-box, plays a key role in coordinating the two binding steps by providing structural flexibility. It is worth mentioning that the impact on the tRNA binding kinetics by mutations in the T-loops, the K-turn, and the Stem III are consistent with the sequence conservation and the in vivo impact on amino acid-mediated transcription read-through using tyrS T-box riboswitches as a model system (*Henkin, 2014*; *Rollins et al., 1997*; *Winkler et al., 2001*). Mutations in the highly conserved T-loop and K-turn region have a more dramatic influence on the tRNA binding kinetics, translating into a larger in vivo impact. While the less conserved Stem III contributes to the stabilization of the anti-terminator conformation in vitro, deletion of it does not appear to significantly impair the tRNA binding process in vitro. Potentially its major function is to create a pause site to coordinate with the co-transcriptional folding of the T-box (*Grundy and Henkin, 2004*; *Zhang and Landick, 2016*).

In conclusion, our study provides a comprehensive kinetic framework for describing tRNA recognition by the T-box riboswitch. The two-step binding process is driven by the specific structural elements of the T-box, and is likely to be kinetically beneficial for efficient, co-transcriptional recognition of the cognate tRNA ligand. Specific T-box structural elements drive the two-step binding process in a modular fashion, that is the 5' and 3' portions of the T-box are responsible for the

first and second binding steps, respectively, with the K-turn region coordinating the two binding steps by allowing structural flexibility, providing a guideline for synthetic biology design of RNA regulatory modules, as well as for the development of new antibiotics using critical T-box structural elements as potential targets. Finally, the *glyQS* T-box riboswitch represents one of the simplest members of this class of riboswitches. Other T-box riboswitches are larger and have additional structural elements and can even appear in tandem arrangements (*Gutiérrez-Preciado et al., 2009*). While the two-step binding kinetics may be common to all T-box riboswitches, it is likely that the process in other T-box riboswitches shows differences modulated by the additional structural elements.

# Materials and methods

## Key resources table

| Reagent type (species) or resource | Designation | Source or reference | Identifiers | Additional information |
|---|---|---|---|---|
| Gene (*Bacillus subtilis*) | *glyQS* T-box riboswitch | PMID: 28531275 | | |
| Gene (*Bacillus subtilis*) | tRNA-Gly | PMID: 9023104 | RRID:SCR_008636 | |
| Strain, strain background (*E. coli*) | DH5α | Thermo Fisher Scientific | | Catalogue # 18265017 |
| Recombinant DNA reagent | pUC19 Vector | New England Biolabs | | Catalogue # N3041S |
| Peptide, recombinant protein | T7 RNA polymerase | PMID: 3684574 | | |
| Peptide, recombinant protein | BsaI | New England Biolabs | | Catalgue #: M0535S |
| Peptide, recombinant protein | RppH | New England Biolabs | | Catalgue #: M0356S |
| Chemical compound, drug | Cyanine5 NHS ester | Lumiprobe | | Catalogue # 13020 |
| Chemical compound, drug | Cyanine3 hydrazide | Lumiprobe | | Catalogue # 13070 |
| Software, algorithm | Origin 5.0 | Microcal Origin | RRID:SCR_002815 | |
| Software, algorithm | ImageJ | ImageJ | RRID:SCR_003070 | |
| Software, algorithm | MATLAB | MATLAB | RRID:SCR_001622 | |
| Software, algorithm | NIS-Elements | Nikon | RRID:SCR_014329 | |

## RNA purification and mutagenesis

RNA transcription was performed in vitro using His$_6$-tagged T7 RNA polymerase using standard protocols (*Milligan et al., 1987*). Cloning, design of a bicistronic DNA template encoding the *B. subtilis glyQS* T-box riboswitch and its cognate tRNA$^{Gly}$, and conditions for in vitro transcription were described before (*Chetnani and Mondragón, 2017*). For the experiments, all the RNAs from the crude transcription reaction were purified on a 7.5% denaturing (8 M Urea) polyacrylamide gel. The RNAs of interest were located on the gel by UV shadowing, the bands were cut out, and the RNAs were eluted into 50 mM sodium acetate (pH 7.0) buffer containing 200 mM potassium chloride by overnight rocking at 4℃. The eluted RNAs were precipitated by adding 3 volumes of cold 100% ethanol and stored overnight at −20℃. The precipitated RNAs were pelleted by centrifugation for 15 min at 20,000 g. The RNA pellets were washed three times in cold 80% ethanol, dried in a Speedvac and re-suspended in water. The concentration was estimated by its absorbance at 260 nm and was kept frozen at −20℃ for long term storage. All mutant constructs were made using a commercial site directed mutagenesis protocol (*Kunkel, 1985*) (Quikchange, Stratagene) and the RNAs were produced and purified by using the same protocol described above. The sequence of all mutant T-box and tRNA constructs were confirmed by sequencing and validated by MFOLD (*Zuker, 2003*) to ensure that the secondary structural elements were not affected by the mutations.

## Fluorescent labelling of the RNA

For smFRET experiments, end labelling of RNA molecules was performed by modifying standard labelling protocols (*Rinaldi et al., 2015*). For 3' end labelling, 50 µg of RNA in 50 µL of reaction volume was incubated with 0.1 M Na-periodate in 0.1 M Na-acetate buffer at pH 5.2 for 90 min in the dark. The reaction was quenched by adding 5 µL of 2.5 M KCl and incubating on ice for 10 min. The resultant insoluble $KIO_4$ was removed by centrifugation at 20,000 g for 30 min and the supernatant was passed through a P6 column (Bio-Rad) to exchange the buffer to 0.1 M HEPES (pH 7.0), 40% DMSO. The RNA was incubated with Cy3 hydrazide (Lumiprobe) dye for 45 min with a final RNA/dye ratio of ~1:200. The RNA was then ethanol precipitated as described above. The precipitated RNA was pelleted by centrifugation for 15 min at 20,000 g, dried in a Speedvac and re-suspended in water. The final RNA solution was passed through a P6 column to remove any residual free dye.

5' end labelling of RNA was performed by N-(3-Dimethylaminopropyl)-N'-ethylcarbodiimide hydrochloride (EDC) – N-hydroxysuccinimide (NHS) coupling through activation of the 5' monophosphate of the RNA by EDC and imidazole (*Rinaldi et al., 2015*). To improve the overall labelling efficiency of this method, a modified approach was used in which the 5' triphosphate of 100 µg of RNA was converted to 5' monophosphate in 100 µL reaction volume by incubating it with 100 units of RNA 5' Pyrophosphohydrolase (NEB) at 37°C for 1 hr. The enzyme was removed by phenol chloroform extraction and the supernatant was passed through a P6 column to exchange the buffer to 10 mM HEPES (pH 7.0), 150 mM NaCl, 10 mM EDTA. This was followed by addition of 12.5 mg of EDC to the RNA solution along with 50 µL of ethylene diamine and 200 µL of 0.1 M imidazole buffer (pH 6.0). The reaction was incubated for 3 hr at 37°C and the RNA was then ethanol precipitated as described above. The resultant RNA pellet was re-suspended in 0.1 M sodium carbonate buffer (pH 8.7) and residual EDC was removed by passing the solution through a P6 column. The resultant RNA solution in 0.1 M sodium carbonate buffer (pH 8.7) was incubated with Cy5 NHS (Lumiprobe) dye for 45 min with a final RNA/dye ratio of ~1:200. The RNA was ethanol precipitated as described above and re-suspended in water. The final RNA solution was passed through a P6 column to remove any residual free dye. The overall labelling efficiency of RNA constructs used in this study varied from 75% to 95%.

## Isothermal titration calorimetry

The thermodynamic parameters associated with binding of tRNA$^{Gly}$ to *glyQS* T-box were determined at 25°C by using an ITC-200 Micro-Calorimeter (MicroCal). Prior to the experiment, the two interacting RNAs (in 50 mM HEPES pH 7.0, 100 mM KCl) were refolded separately by first heating for 3 min at 90° C followed by incubation on ice for 2 min. At this point, $MgCl_2$ was added to attain a final concentration of either 1 mM or 10 mM. The RNA solution was then heated to 50° C for 10 min and 37° C for 30 min followed by cooling to 25°C. The refolded RNA was then concentrated using a 10 kDa cutoff Amicon filter and washed three times with the final ITC buffer (50 mM HEPES pH 7.0, 100 mM KCl and 10 mM $MgCl_2$ or 1 mM $MgCl_2$). For testing the binding of T-box with unlabeled and 5' Cy5 labelled tRNA in 10 mM $MgCl_2$ buffer, the sample cell was filled with 6.6 µM and 11 µM T-box respectively and the corresponding concentration of tRNA$^{Gly}$ in the syringe was 111 µM and 152 µM. For the ITC experiment in 1 mM $MgCl_2$ buffer, the cell was filled with T-box at a concentration of 6.7 µM and the syringe concentration of unlabeled tRNA$^{Gly}$ was 72 µM. For each ITC experiment, the titration was carried out by stepwise (2 µL) injection of tRNA$^{Gly}$ from a stirred syringe (1000 rev/min) into the sample cell. Successive injections were spaced by 150 s and values for the change in enthalpy ($\Delta H_b$), association constant ($K_b$), change in enthalpy ($\Delta H_b$) and stoichiometry (n) were determined by nonlinear least-squares fitting of the data using Origin 5.0 software (OriginLab).

## Electrophoretic mobility shift assay

The T-box and tRNA were denatured and refolded using the same protocol as described for the ITC and smFRET experiments. For binding of tRNA, 5 µL of 4 µM of folded T-box and 5 µL of 2 µM folded tRNA samples were mixed together for 30 min in final buffer containing 50 mM HEPES pH 7.0, 100 mM KCl and 15 mM $MgCl_2$. The folded T-box or T-box +tRNA mixture were loaded on to 6% native polyacrylamide gel containing 15 mM $MgCl_2$ and the gel was run at room temperature for 2–2.5 hr. RNAs were stained by SYBR$^{TM}$ green RNA staining dye (Invitrogen). The gel was imaged

using ChemiDoc (Bio-rad) in SYBR green channel (for unlabeled T-box constructs), Cy5 channel (for tRNA-Cy5), or Cy3 channel (for labeled T-box constructs). Images were processed and analyzed by ImageJ (*Schneider et al., 2012*).

## Fluorophore conjugation of DNA oligos

DNA oligos that hybridize to 5' extension and 3' extension of the T-box construct were purchased from Integrated DNA Technologies with an amine modification at the 5' end and 3' end respectively. 13.5 μL of 100 μM DNA oligo was mixed with 1.5 μL of 1 M NaHCO$_3$ (pH 8.6). 25 μg of NHS conjugated fluorophore (Cy3 or Cy5) was dissolved with 0.5 μL DMSO and mixed with the DNA oligo solution. The mixture was incubated at 37$^{°C}$ overnight. 1.67 μL of 3 M NaOAc and 50 μL of pure ethanol was added to the mixture to precipitate the conjugated DNA oligo overnight at −20°C. The precipitated DNA oligo was pelleted by centrifugation for 30 min at 21000 g and re-suspended with 40 μL water. The DNA solution was passed through a P6 column to remove any residue free dye and salt. The overall fluorophore labeling efficiency is ~60%.

## smFRET measurements

Slides containing microfluidic channels were prepared as previously described (*Blanchard et al., 2004*). Slides and coverslips were coated with a mixture of poly-ethylene glycol (PEG, Mw = 500,000) and PEG-biotin (Mw = 500,000) according to previously published protocol (*Blanchard et al., 2004*). The T-box and tRNA were denatured and refolded using the same protocol described for the ITC experiments with the final buffer containing 50 mM HEPES pH 7.0, 100 mM KCl, and 15 mM MgCl$_2$. T-box RNA was hybridized to the biotinylated DNA oligo at the extension during refolding, and immobilized via biotin-streptavidin interactions to the surface. tRNA was diluted in imaging buffer (50 mM HEPES pH 7.0, 100 mM KCl, 15 mM MgCl$_2$, 5 mM protocatechuic acid (PCA) (Sigma), 160 nM protocatechuate-3,4-dioxygenase (PCD) (Sigma), and 2 mM Trolox (Sigma)) and flowed into the microfluidic channels. smFRET measurements were performed with an objective based total internal reflection fluorescence (TIRF) microscope based on a Nikon Ti-E with 100X NA 1.49 CFI HP TIRF objective (Nikon). A 561 nm laser (Coherent Obis at a power density of 4.07 × 10$^5$ W/cm$^2$) was used for the FRET measurement. A 647 nm laser (Cobolt MLD at a power density of 5.88 × 10$^5$ W/cm$^2$) was used for direct excitation of Cy5 to check the presence of the acceptor. Emissions from both donor and acceptor were passed through an emission splitter (Opto-Split III, Cairn), and collected at different locations on an EMCCD (iXon Ultra 888, Andor). 1500 frames of time-lapse images were taken with 100 ms exposure time. Each independent measurement in our study should be considered as a technical replicate. The biological samples (i.e. the in vitro transcribed tRNAs and T-boxes) were generated once. In each measurement, T-box and tRNA were folded, smFRET images were recorded, and analysis were performed independently.

## Lifetime analysis

Individual spots were picked from maximum intensity projection of Cy5 emission channel (*Figure 2—figure supplement 1*) using NIS Elements software, that is only the pixels with the highest intensity value of the same XY coordinates in time-lapse images are displayed in the maximum intensity projection image. Fluorescent intensity trajectories of these spots were generated from Cy3 and Cy5 channels of time-lapse images, and corrected for baseline and bleed-through in MATLAB as previously described (*Fei et al., 2008*). FRET traces were generated by calculating $I_{Cy5}$ / ($I_{Cy5}$+$I_{Cy3}$) at each time point from the intensity trajectories. smFRET traces were idealized by fitting with a Hidden Markov Model using vbFRET (*Bronson et al., 2009*). Dwell time of each FRET state before transition to another FRET state of individual traces was extracted from the idealized traces and the dwell time histograms were fit with exponential decay by Origin 7.0 (OriginLab) (*Figure 2—figure supplement 2*). In most of the cases, dwell time histograms fit well with a single exponential decay ($A.\exp(-t/t_0) + y_0$). In a few cases, in which mixed populations with different lifetimes were expected, dwell time histograms were fit with a double exponential decay ($A_1.\exp(-t/t_1) + A_2.\exp(-t/t_2) + y_0$) and the population-weighted average lifetime was calculated by ($A_1.t_1 + A_2.t_2$) / ($A_1 + A_2$). The latter cases are explicitly mentioned in the text.

## Determination of kinetic parameters

### (I) $k_{-1}$

For all T-box constructs, $k_{-1}$ was estimated from the dwell time of the partially bound state (0.4 FRET state of T-box$_{182}$, T-box$_{C56U}$ and T-box$_{SIII-\Delta4bp}$) of tRNA$^{\Delta NCCA}$-Cy5 (*Figure 2—figure supplement 2*). The dwell time of the partially bound state was fit with a single exponential decay. The lifetime of 0.4 FRET state ($\tau_{0.4-0}$) was extracted and $k_{-1}$ was calculated using the following equation:

$$k_{-1} = \frac{1}{\tau_{0.4-0}}$$

### (II) $k_1$

We noticed that the lifetime of the unbound state ($\tau_{0-0.4}$) of tRNA$^{\Delta NCCA}$-Cy5 with Tbox$_{182}$-Cy3(3') was 31.3 ± 5.3 s, while the average smFRET trace length ($\tau_{trace}$) was 70 ± 7 s under 4.07 × 10$^5$ W/cm$^2$ 561 nm laser (*Figure 2—figure supplement 2*), limited by Cy3 photobleaching. Therefore, there was a possibility of underestimating the lifetime of the unbound state, or overestimating $k_1$. We hence characterized the photobleaching effect on unbound state lifetime analysis using reduced 561 nm laser power (1.96 × 10$^5$ W/cm$^2$). $\tau_{0-0.4}$ was calculated to be 46 ± 4 s and $\tau_{trace}$ was 103.0 ± 0.6 s. To correct for the photobleaching effect, we estimated $k_1$, using the following equation (*Bartley et al., 2003*; *Fei et al., 2009*):

$$k_1 = \left( \frac{1}{\tau_{0-0.4}} - \frac{1}{\tau_{trace}} \right) \times \frac{1}{[tRNA]}$$

where [tRNA] represents the concentration of tRNA$^{\Delta NCCA}$-Cy5. $k_1$ values were consistent using both high laser power ((5.8 ± 0.2) x 10$^5$ M$^{-1}$s$^{-1}$) and low laser power ((4.1 ± 0.6) x 10$^5$ M$^{-1}$s$^{-1}$). In *Figure 6E*, we reported the mean ±S.D. of $k_1$ obtained with both high and low laser powers.

We also measured $\tau_{off}$ in the flow experiment and it was determined by the dwell time between the injection time and detection of the first binding event (*Figure 2—figure supplement 4C*). $k_1$ calculated for tRNA$^{Gly}$ in the flow experiment was consistent with the value for tRNA$^{\Delta NCCA}$ in the steady-state experiment. We therefore used $k_1$ of tRNA$^{\Delta NCCA}$-Cy5 binding to the T-box to approximate the rate constant for the first binding step in all T-box constructs.

For some T-box mutants (specifically, T-box$_{C56U}$-Cy3 and T-box$_{\Delta KT}$-Cy3), binding of tRNA-Cy5 became less efficient compared to the T-box$_{182}$-Cy3(3'), already a hint of a deficient first binding step. In this case, estimation of $k_1$ from the selected FRET traces will highly bias for the molecules that bind relatively fast within the photobleaching time of Cy3, therefore significantly overestimating $k_1$. We therefore applied a different approach to estimate $k_1$ for the T-box mutants. We counted trace percentage of the T-box mutants (number of FRET traces divided by the number of Cy3 spots) and normalized it to the trace percentage of T-box$_{182}$ with tRNA$^{\Delta NCCA}$-Cy5 to get the normalized trace percentage (*f*) (*Figure 6—figure supplement 4*).

We performed Gillespie simulations (*Gillespie, 1976*; *Gillespie, 1977*; *Gillespie, 2007*) on the observed signal as a function of $k_1$ and $k_{-1}$ during the first binding step. The simulated signal mimicked the experiment and imaging processing. As our FRET traces were selected from the maximum intensity projection of the acceptor signal (*Figure 2—figure supplement 1*) regardless of the lifetime of the bound state, as long as there was a binding event during the imaging time window (70 s considering the photobleaching time of the donor dye), it would generate a signal (as a 'footprint') that showed in the output image of maximum projection (*Figure 6—figure supplement 5A*). For each parameter set, 2000 binding trajectories were simulated, and the fraction of the molecules that generated signal in the maximum intensity projection (referred to as '*f*') within the 70 s time window was calculated. The simulation shows that *f* is insensitive to $k_{-1}$, at least within the $k_{-1}$ we observed for the various T-box constructs (*Figure 6—figure supplement 5B*), and that the dependence of *f* on $k_1$ is best described as

$$f = f_0 - ae^{bk_1}$$

where $f_0$ is the binding fraction at saturation, and *a* and *b* are constants (*Figure 6—figure supplement 5B*). By fitting $\ln(f_0-f)$ *vs.* $k_1$ with a linear function (*Figure 6—figure supplement 5C*) we estimated $k_1$ as

$$k_1 = \frac{0.45 - \ln(1.557 - f)}{72.09}$$

The standard deviations reported for $k_1$ were calculated by an error propagation function considering the standard deviation of $f$.

### (III) $k_2$ and $k_{-2}$

Even though we observed that smFRET traces of tRNA$^{Gly}$ with T-box$_{182}$, T-box$_{C56U}$ and T-box$_{SIII-\Delta4bp}$ exhibit real-time binding, the majority of the traces show single-step binding from unbound to fully bound state while only a few traces sample the partially unbound state within 0.1 s (*Figure 2F*, *Figure 2—figure supplement 4B*,*Figure 6—figure supplements 1A* and *2A*). For this reason, our calculation of $k_2$ is limited by the imaging time resolution and we estimate the transition from 0.4 to 0.7 is rapid with a $k_2$ larger than 10 s$^{-1}$.

We can also detect transition from 0.7 FRET to 0.4 FRET, and calculate the lifetime of 0.7 FRET to 0.4 FRET ($\tau_{0.7-0.4}$) (*Figure 2—figure supplement 2*). However, the fraction of traces showing this transition is small. Since most of the traces demonstrate long time 0.7 FRET, we reason that $k_{-2}$ should be very small and the second binding step for these three constructs is close to irreversible.

### (IV) Kinetic model for T-box$_{\Delta KT}$

The kinetic model for tRNA binding to the K-turn mutant T-box$_{\Delta KT}$ is illustrated in *Figure 6—figure supplement 3C*, with kinetic parameters marked. Specifically, the partially bound state contains a relaxed conformation and a bent conformation at the original K-turn region. The interconversion rates between 0.2, 0.4 and 0.7 FRET states were determined by the lifetime of the each FRET state before transitioning to the other FRET state. To compare better the kinetic parameters with other T-box constructs, we calculated the apparent $k_2$ considering both pathways (0.2→0.7 and 0.2→0.4→0.7) (*Figure 6—figure supplement 3C*) as below:

$$k_{2\_app} = \left( p_1 \frac{k_1' k_2'}{(k_1' + k_{-1}')} + p_2 k_2 \right)$$

Where $p_1$ and $p_2$ are the probabilities of the 0.2→0.4→0.7 and 0.2→0.7 pathways respectively. And we calculated the apparent $k_{-2}$ considering both pathways (0.7→0.2 and 0.7→0.4) (*Figure 6—figure supplement 3C*) as shown below:

$$k_{-2\_app} = \left( p_1 k_{-2}' + p_2 k_{-2} \right)$$

Where $p_1$ and $p_2$ are the probabilities of the 0.7→0.4 and 0.7→0.2 pathways, respectively. To compare with the $k_{-1}$ of other T-box constructs, the apparent $k_{-1\_app}$ of T-box$_{\Delta KT}$ was estimated by considering both 0.4 and 0.2 FRET states.

## Acknowledgements

This work was funded by a grant to JF and AM by the Chicago Biomedical Consortium with support from the Searle Funds at the Chicago Community Trust. We thank E Heideman for the preparation of slides for all imaging experiments and managing the Fei laboratory.

## Additional information

### Funding

| Funder | Author |
| --- | --- |
| Chicago Community Trust | Alfonso Mondragón<br>Jingyi Fei |

The funders had no role in study design, data collection and interpretation, or the decision to submit the work for publication.

## Author contributions

Jiacheng Zhang, Designed the research, Performed single molecule FRET experiments, Data analysis, Wrote and revised the manuscript; Bhaskar Chetnani, Designed the research, Prepared samples and performed ITC experiments, Wrote and revised the manuscript; Eric D Cormack, Helped with smFRET imaging and data analysis; Dulce Alonso, Sample preparation and ITC experiments; Wei Liu, Helped with data analysis; Alfonso Mondragón, Designed the research, Wrote and revised the manuscript; Jingyi Fei, Designed the research, Helped with data analysis, Wrote and revised the manuscript

## Author ORCIDs

Jiacheng Zhang (iD) http://orcid.org/0000-0002-8671-5888
Alfonso Mondragón (iD) https://orcid.org/0000-0002-0423-6323
Jingyi Fei (iD) http://orcid.org/0000-0002-9775-3820

## Decision letter and Author response

Decision letter https://doi.org/10.7554/eLife.39518.044
Author response https://doi.org/10.7554/eLife.39518.045

# Additional files

## Supplementary files

• Supplementary file 1. Secondary structure diagram of the constructs used for the smFRET studies. The diagrams show all the constructs used for the experiments. The 5' extensions added to attach the T-box constructs to the surface are shown in red. The $Tbox_{182}$ construct was made with only a 5'extension and also with extensions at both the 5' and 3' ends (red).
DOI: https://doi.org/10.7554/eLife.39518.041

• Transparent reporting form
DOI: https://doi.org/10.7554/eLife.39518.042

## Data availability

All data generated or analysed during this study are included in the manuscript and supporting files. smFRET trajectories of each data sets included in the manuscript are available in Source Data.

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
