## [Decision Letter]

Thank you for submitting your article "Specific structural elements of the T-box riboswitch drive the two-step binding of the tRNA ligand" for consideration by *eLife*. Your article has been reviewed by Gisela Storz serving as both Reviewing and Senior Editor, a Reviewing Editor, and three reviewers. The following individuals involved in review of your submission have agreed to reveal their identity: Adrian Ferre-D'Amare (Reviewer #2).

The reviewers have discussed the reviews with one another and the Reviewing Editor has drafted this decision to help you prepare a revised submission.

Summary:

The manuscript by Zhang et al., describes the results of smFRET studies on the recognition of uncharged tRNA by the T-box riboswitch. These are widespread bacterial riboregulators that employ the charge state of tRNA as a proxy for the intracellular concentration of specific amino acids. Previous genetic, biochemical and structural studies have elucidated the nature of the RNA-RNA interactions that give rise to specific interactions between the T-box and a cognate tRNA. However, the kinetics of the association have remained largely unexplored. In this manuscript, the authors use smFRET to characterize the association of a model T-box with its cognate tRNA. By judicious combination of labeling schemes as well as mutant T-boxes, the authors derive a two-step binding model and provide, for the first time, a quantitative kinetic framework for the association of the two RNAs. The T-box-tRNA interaction is an outstanding model system to elucidate how two non-coding RNAs associate specifically, making this work not only of interest to those who study riboswitches, but to the larger community of scientists who study non-coding RNAs.

Essential revisions:

Overall, the manuscript is well written and easy to follow, and the interpretations are sound. The highlight of the study is the capture and mechanistic interpretation of the partially bound states, which reflects the strength of single molecule techniques.

1) As the authors note, a superficially similar paper from the Nikonowicz and Walter labs analyzing T-box-tRNA association was published recently. However, the labeling scheme used by those authors precludes the analysis of T-box conformational accommodation. This is an important distinction, and the current study complements the recently published work from the Nikonowicz and Walter labs in important ways. However, the description of the work by Suddala et al., should be improved, Zhang et al., should discuss more clearly what insights they have obtained and how they extend and complement the experiments from the Nikonowicz and Walter labs. In particular, Suddala et al., carried out colocalization experiments, which are extremely powerful in detecting binding events that do not result in FRET. In addition, they observed Mg^2+^ dependence on the switching, providing additional evidence that the K-turn switch may be metal dependent. They also observed three different off rates (k^fast^, k^medium^, and k^slow^). It is unclear how these three rates relate to the two states proposed here. Finally, Suddala et al., provided data that EFTu does not affect binding kinetics, which should likely be cited in this paper.

2) For broad interest to the bacterial gene regulation and riboswitch communities, the authors should expand on the in vivo consequences of the in vitro kinetics. Specifically:

- The authors at their own discretion may consider referencing the biological consequences of the mutations that are studied using the single molecule approach (if that information is already available)? For example, what happens in bacteria as a result of K-turn mutation, mutation C56, or stem III truncation? If this information is already in hand, it could help tie together the kinetics results with in vivo consequences for riboswitch regulation.

- The authors could provide a much more detailed link between their kinetic mechanism and the regulatory consequences given observed rates of transcription and pausing. The authors can potentially put numbers to many of these events. This has been done to some extent in the work by Suddala et al. It would be interesting to see how things change with the kinetics observed here.

- In general, the authors may undersell the two-step mechanism a bit. In what types of other systems (e.g., Cas9? Ago?) have two step binding mechanisms been observed? Is this potentially a general strategy for promoting fidelity? Would incorporating a proofreading/fidelity step be useful for thinking about biotechnology applications?

3) The labeling scheme employed in this work places a dye at the 5'-terminus of tRNA. While qualitative, previous biochemical work (Zhang and Ferre-D'Amare, (2014) suggested intimate coaxial stacking between the antiterminator element of the T-box and the acceptor stem of tRNA. Thus, an important control experiment is to show that placing a dye at the 5'-terminus of tRNA does not impact tRNA-T-box association, or to determine what the effect is. An ITC experiment (like the one authors’ report, for other purposes, in Figure 3—figure supplement 2) would be pretty clear in this regard.

4) The experiment establishing the kink-turn as the hinge point for the conformational change between the partially bound and full complex is on the weak side. The authors claim that the absence of a FRET change in their experiment is evidence that the kink-turn must be the point where flexing occurs. An alternative interpretation of this experiment could be that this just shows that the tRNA is not changing position relative to the anchor point during this transition, rather than pinpointing to where the T-box is bending to form the full complex. An experiment that gives a positive result upon kink-turn bending and full complex formation would be stronger evidence for the claim that the kink-turn is the main source of conformational change.

5) The authors conclude from their experiments that tRNA binding is a two-step process where the Stem-I interaction occurs first followed by the anti-terminator interaction. This is likely true in vivo where Stem-I emerges from the polymerase first. However, in their system the possibility exists that the anti-terminator interaction occurs first followed by Stem-I. Would the authors be able to discern FRET traces where this occurs using their set-up? Are transitions besides the.4 to.7 state observed? If the anti-terminator is never found to be bound first, can the authors provide a reason?

6) Subsection “Binding of cognate tRNA by the glyQS T-box results in two distinct FRET states”, it is unclear why some molecules are transitioning here, and others are not. In the recently published T-box paper from the Walter lab, Walter et al., stressed that their T-box was purified under native conditions. While in this work the RNA was denatured. How was the RNA refolded? Have the authors run a native gel of their RNA to detect possible folding heterogeneity that could explain some of these results?

7) Subsection “Determination of kinetic parameters”, can the authors provide additional information (or a reference or a simulation) for adjusting the fitted kinetic parameter based on the percentage of data observed? Does this really scale strictly as a percentage? How does the error in the fit scale?

---

## [Author Response]

Essential revisions:Overall, the manuscript is well written and easy to follow, and the interpretations are sound. The highlight of the study is the capture and mechanistic interpretation of the partially bound states, which reflects the strength of single molecule techniques.

We are glad that the reviewers found that our manuscript is well written and easy to follow, the interpretations are sound, and the conclusions are of interest to the broader scientific community.

1) As the authors note, a superficially similar paper from the Nikonowicz and Walter labs analyzing T-box-tRNA association was published recently. However, the labeling scheme used by those authors precludes the analysis of T-box conformational accommodation. This is an important distinction, and the current study complements the recently published work from the Nikonowicz and Walter labs in important ways. However, the description of the work by Suddala et al.,. should be improved, Zhang et al., should discuss more clearly what insights they have obtained and how they extend and complement the experiments from the Nikonowicz and Walter labs. In particular, Suddala et al., carried out colocalization experiments, which are extremely powerful in detecting binding events that do not result in FRET. In addition, they observed Mg^2+^ dependence on the switching, providing additional evidence that the K-turn switch may be metal dependent. They also observed three different off rates (k^fast^, k^medium^, and k^slow^). It is unclear how these three rates relate to the two states proposed here. Finally, Suddala et al., provided data that EFTu does not affect binding kinetics, which should likely be cited in this paper.

In the revised manuscript the comparison between our study and the study from the Nikonowicz and Walter laboratories is more extensive and complete. It emphasizes the experiments and conclusions in their study that are different from our studies. To a large extent, both studies are highly complementary and address non-overlapping aspects of T-box/tRNA binding kinetics. We made the following specific changes, as suggested by the reviewers, and included them in the Results section and Discussion section.

We now mention in the discussion that Suddala et al., used colocalized signals from the bound tRNA and the T-box to study the tRNA binding kinetics. While we did not use colocalization as a way to study tRNA binding, our choice of tRNA labeling positions makes the FRET value high enough (0.4 at least) to use the FRET signal reliably to report on binding. In fact, our k_on_ and k_off_ estimates from the partially bound state are highly consistent with the numbers reported by Suddala et al., providing further validation for using FRET signal to report binding.

We now mention the Mg-dependent conformational change reported by Suddala et al., when we discuss the K-turn mutant.

“In addition, such conformational changes mediated by the K-turn region are Mg^2+^ dependent, as suggested by Suddala et al. (Suddala et al., 2018).”

In addition, in Figure 1—figure supplement 2 we include ITC binding data done at 1mM MgCl2 concentration that shows clearly that tRNA binding at this Mg concentration is completely abolished. This shows that tRNA binding by the T-box is Mg dependent and the T-box is an Mg-dependent switch.

The lifetime of the fully bound state is consistent with the k_off_^medium^ in the study by Suddala et al. As we did not perform imaging with low time resolution (e.g. 20 s as in the work by Suddala et al.), we cannot fully distinguish k_off_^medium^ and k_off_^slow^ in our results. It is possible that the fully bound state contains two populations with tRNA dissociating with different rates. However, in our model the key conclusion is that the fully bound state is a relatively stable complex with a lifetime of at least 24 s, and probably longer, which is long enough to be functionally important as it allows transcription of the terminator sequence. We include the comparison of the lifetimes obtained in both studies in the Discussion section:

“By using colocalized signals from the bound tRNA and the immobilized T-box, Suddala et al. (Suddala et al., 2018) uncovered two binding states distinguished by different dissociation rates of the tRNA, aided by using a Stem I-only mutant that cannot interact with the NCCA end of the tRNA. Specifically, in the model of Suddala et al. (Suddala et al., 2018) binding of the anticodon of the uncharged tRNA results in a relatively unstable state (with a lifetime of ~4-5 s), consistent with the partially bound state in our model, while binding both the anticodon and the NCCA end of the tRNA results in a stable state, consistent with the fully bound state in our model.”

and

“Finally, the fully bound state has a lifetime of at least 24 s, allowing RNAP to elongate more than 1,000 nts before the tRNA dissociates. Therefore, the second binding step in the WT T-box with uncharged tRNA^Gly^ is close to irreversible in such biological setting. […] Nevertheless, both potential configurations in the fully bound state are stable enough to allow transcription read-through.”

We have now cited the paper in the introduction when mentioned EF-Tu.

*2) For broad interest to the bacterial gene regulation and riboswitch communities, the authors should expand on the* in vivo *consequences of the* in vitro *kinetics. Specifically:*

*- The authors at their own discretion may consider referencing the biological consequences of the mutations that are studied using the single molecule approach (if that information is already available)? For example, what happens in bacteria as a result of K-turn mutation, mutation C56, or stem III truncation? If this information is already in hand, it could help tie together the kinetics results with* in vivo *consequences for riboswitch regulation.*

We now cite work from the Henkin laboratory on in vivo characterization of tyrS T-box riboswitches carrying mutations in the T-loop and the K-turn regions. While these in vivo studies are on a different T-box, these two regions are highly conserved, therefore it is possible to expect similar in vivo functional attenuation by glyQS T-box riboswitches carrying equivalent mutations. As we indicated previously in the manuscript, the Stem III mutant shows a minor impact on tRNA binding kinetics and hence we speculate that the in vivo relevance of Stem III is related to the promotion of transcriptional pausing during T-box co-transcriptional folding. Unlike the other two regions, Stem III is not highly conserved. We revised the manuscript as below:

“It is worth mentioning that the impact on the tRNA binding kinetics by mutations in the T-loops, the K-turn, and the Stem III are consistent with the sequence conservation and the in vivo impact on amino acid-mediated transcription read-through using tyrS T-box riboswitches as a model system (Henkin, 2014; Rollins et al., 1997; Winkler et al., 2001). Mutations in the highly conserved T-loop and K-turn region have a more dramatic influence on the tRNA binding kinetics, translating into large in vivo impact. While the less conserved Stem III contributes to the stabilization of the anti-terminator conformation in vitro, deletion of it does not appear to significantly impair the tRNA binding process in vitro. Potentially its major function is to create a pause site to coordinate with the co-transcriptional folding of the T-box (Grundy and Henkin, 2004; Zhang and Landick, 2016).”

- The authors could provide a much more detailed link between their kinetic mechanism and the regulatory consequences given observed rates of transcription and pausing. The authors can potentially put numbers to many of these events. This has been done to some extent in the work by Suddala et al. It would be interesting to see how things change with the kinetics observed here.

Previously we mentioned briefly in the Discussion section the potential impact of the tRNA binding kinetics in the co-transcriptional setting. In the revised manuscript, we have significantly expanded such discussion as follows:

“T-box riboswitches fold and function co-transcriptionally in the cell, and hence we propose that a two-step kinetic model is kinetically beneficial during co-transcriptional folding of the T-box and sensing of the tRNA ligand. […] Finally, the fully bound state has a lifetime of at least 24 s, allowing RNAP to elongate more than 1,000 nts before the tRNA dissociates. Therefore, the second binding step in the WT T-box with uncharged tRNA^Gly^ is close to irreversible in such biological setting.”

- In general, the authors may undersell the two-step mechanism a bit. In what types of other systems (e.g., Cas9? Ago?) have two step binding mechanisms been observed? Is this potentially a general strategy for promoting fidelity? Would incorporating a proofreading/fidelity step be useful for thinking about biotechnology applications?

We thank the reviewers for this suggestion. We have now included discussion on two-step mechanisms utilized in other biological process as follows.

“Two-step binding mechanisms have been observed in a variety of protein or nucleic acid mediated biological process, including T-cell receptor (TCR) recognition of the major histocompatibility complex (MHC) presenting peptides, where TCRs scan the MHC scaffold first, followed by sensing of specific MHC-presenting peptides (Wu et al., 2002); interaction of the signal recognition particle (SRP) receptor with the membrane, where SRP receptors interact with the membrane in a dynamic mode followed by an SRP-induced conformational transition into a stable binding mode (Hwang Fu et al., 2017), DNA interrogation by CRISPR Cas9-crRNA, where Cas9-crRNA recognizes the protospacer adjacent motif (PAM) on the target DNA followed by sensing of the spacer sequence and triggering R-loop formation (Sternberg et al., 2014); and RNA-induced silencing complexes (RISCs) binding to their mRNA targets, where dynamic sampling of the “sub-seed” region occurs before targeting stably across the full seed region (Chandradoss et al., 2015; Herzog and Ameres, 2015; Salomon et al., 2015, 2016).

In all of these cases, a two-step binding mechanism provides a good balance between sensitivity and specificity.”

3) The labeling scheme employed in this work places a dye at the 5'-terminus of tRNA. While qualitative, previous biochemical work (Zhang and Ferre-D'Amare, (2014)) suggested intimate coaxial stacking between the antiterminator element of the T-box and the acceptor stem of tRNA. Thus, an important control experiment is to show that placing a dye at the 5'-terminus of tRNA does not impact tRNA-T-box association, or to determine what the effect is. An ITC experiment (like the one authors’ report, for other purposes, in Figure 3—figure supplement 2) would be pretty clear in this regard.

We thank the reviewers for this critical comment. We have performed ITC experiments on the binding of Cy5 labeled tRNA on the 5’ end. This is not an easy experiment as it requires large amounts of labeled tRNA, but we succeeded in making it work. A revised figure (Figure 1—figure supplement 2) showing the new ITC result is now part of the manuscript with the experimental details in the figure caption and in the Materials and methods section. As expected, Cy5 placed at the 5’ end the tRNA modestly affects binding, decreasing the binding constant by ~4 fold and suggesting that the fully bound state should be even more stable than measured. As the Cy5 dye is placed close to the NCCA end of the tRNA, the effect of the label should be limited to the second binding step. The fact that the k_1_ and k_-1_ of the first binding step in our study are highly consistent with the reported values by Suddala et al., also suggests that the label only affects the second binding step. Overall, the lifetime of the fully bound state is, as we indicate in the manuscript, underestimated not only due to fluorophore photobleaching, but also due to the 5’ label. In this sense, the minor pitfall introduced by the label does not change the overall conclusions. We revised the Discussion section of the manuscript to include this.

“Without the NCCA interaction, the binding of tRNA is unstable, with a mean lifetime of ~4 s, whereas with interactions both with the anticodon and the NCCA end, the binding of tRNA is very stable, with a mean lifetime > 24 s. It should be noted that the latter lifetime is likely to be much longer for two reasons: first, the measurement is limited by fluorophore photobleaching, which limits long measurements in our experiments; second, the Cy5 label placed at the 5’ end of the tRNA reduces the overall binding affinity by ~ 4-fold (Figure 1—figure supplement 2), likely because the fluorophore impairs the NCCA/t-box interactions to some extent.”

4) The experiment establishing the kink-turn as the hinge point for the conformational change between the partially bound and full complex is on the weak side. The authors claim that the absence of a FRET change in their experiment is evidence that the kink-turn must be the point where flexing occurs. An alternative interpretation of this experiment could be that this just shows that the tRNA is not changing position relative to the anchor point during this transition, rather than pinpointing to where the T-box is bending to form the full complex. An experiment that gives a positive result upon kink-turn bending and full complex formation would be stronger evidence for the claim that the kink-turn is the main source of conformational change.

We thank the reviewers for pointing out this premature conclusion. We agree with the reviewers that insignificant FRET change between the 5’ end of the tRNA and the base of Stem I is not sufficient to prove that the hinge is at the K-turn region. In order to provide positive evidence for this conclusion, the most straightforward way would be visualize the structures of the Tbox/tRNA complex at the partially bound and fully bound states, which is beyond the scope of the current study. However, as the reviewers suggest, our data show the 5’ end of the tRNA maintains its distance, or configuration, relative to the base of the Stem I during the second binding step. In addition, the kinetics of the k-turn mutant indicate that the flexibility of the k-turn is very important for the transition from the partially bound to the fully bound state. Therefore, we decided to revise the change the subtitle of the related section and also the Discussion section in the manuscript to simply mentioning the possibility that the k-turn is likely to be the hinge region. the new subtitle reads:

“The NCCA end of the uncharged tRNA maintains its relative position to the K-turn region during the second binding step”

While the Discussion section was changed as follows:

“Based on the known structures of Stem I in complex with tRNA^Gly^ (Grigg and Ke, 2013; Zhang and Ferre-D'Amare, 2013), it was hypothesized that the intra-T-box conformational change is likely to involve the K-turn region, which sits at the junction between the 5’ and 3’ portions of the T-box. Our observation that the NCCA end of the uncharged tRNA moves towards the anti-terminator stem (revealed by the T-box_182_-Cy3(3’) and tRNA^Gly^-Cy5 FRET), but maintains its relative position to the K-turn region (revealed by the T-box_182_-Cy3(5’) and tRNA^Gly^-Cy5 FRET) during the second binding step is consistent with the role of the K-turn region as the hinge of the intra-T-box conformational change.”

*5) The authors conclude from their experiments that tRNA binding is a two-step process where the Stem-I interaction occurs first followed by the anti-terminator interaction. This is likely true* in vivo *where Stem-I emerges from the polymerase first. However, in their system the possibility exists that the anti-terminator interaction occurs first followed by Stem-I. Would the authors be able to discern FRET traces where this occurs using their set-up? Are transitions besides the.4 to.7 state observed? If the anti-terminator is never found to be bound first, can the authors provide a reason?*

We concluded that the two-step binding process samples the 0.4 FRET state first for the following two reasons: (1) due to the imaging time resolution, most of the traces showed singlestep FRET increases from zero to 0.7 during the flow experiment, where we can capture the realtime binding. Nevertheless, we still see traces showing sampling of the 0.4 FRET state before the 0.7 FRET. (2) Perhaps a stronger argument is that we did not observe any transient binding of tRNA^Tyr^, which has an intact and accessible NCCA, but not the correct anticodon. The latter observation supports the model that anticodon binding needs to occur before establishment of the NCCA interaction. We modified the manuscript to make the point more implicit.

“Collectively, our results suggest a two-step binding model involving the separate establishment of the interactions with the anticodon and the NCCA. The fact that tRNA^Tyr^, which has a mismatched anticodon, but contains an intact NCCA 3’ end, does not show any binding activity suggest that interactions with the anticodon precede the interactions with the NCCA end of the tRNA and are necessary for the establishment of the NCCA contacts.”

6) Subsection “Binding of cognate tRNA by the glyQS T-box results in two distinct FRET states”, it is unclear why some molecules are transitioning here, and others are not. In the recently published T-box paper from the Walter lab, Walter et al., stressed that their T-box was purified under native conditions. While in this work the RNA was denatured. How was the RNA refolded? Have the authors run a native gel of their RNA to detect possible folding heterogeneity that could explain some of these results?

We have added native gel electrophoresis analysis of the T-box folding as a supplement to Figure 1 (Figure 1—figure supplement 1). There is some folding heterogeneity in the T-box, but not in the tRNA, which is fully folded. The T-box constructs, including the WT and the mutants, all consistently show a 60-75% folding efficiency. Nevertheless, when we mix folded T-box with tRNA using a gel shift assay we do not observe tRNA binding to the misfolded/unfolded fraction or to the T-box that is not put through the refolding process. Therefore, we do not think the unfolded/misfolded fraction affects our smFRET measurement or data interpretation. We do not have a good explanation for the ~10% subpopulation that shows transient back transition into the partially bound state yet. Therefore, we do not provide any biological meaning for that small subpopulation.

7) Subsection “Determination of kinetic parameters”, can the authors provide additional information (or a reference or a simulation) for adjusting the fitted kinetic parameter based on the percentage of data observed? Does this really scale strictly as a percentage? How does the error in the fit scale?

We thank the reviewers for pointing this out and helping make our analysis more rigorous. We performed a Gillespie simulation on the observed signal as a function of k_1_ and k_-1_ during the first binding step, which is described in detail in the Materials and methods section. The observed signal in the simulation mimicked the experiment and imaging processing. The simulation shows that the fraction of molecules generating a binding signal is insensitive to k_-1_. The fraction does not scale linearly with apparent k_1_ within a range spanning 6 orders of magnitude (10^-7^ to 10^-1^ s^-1^); however, within the range of our experimental results, a linear relationship is still a good approximation (Pearson’s R = 0.99, see figure below). Nevertheless, we decided to modify the k_1_ calculation with a better fit according to the simulation, as described in the Materials and methods section and add a new figure, Figure 5—figure supplement 5. The modified calculation gives very similar results to the previous calculation within the linear approximation, and therefore all conclusions regarding the effects of the mutations still hold.